# Masked Diffusion Models are Secretly Time-Agnostic Masked Models and Exploit Inaccurate Categorical Sampling

**Kaiwen Zheng**[1]*, **Yongxin Chen**[2], **Hanzi Mao**[2], **Ming-Yu Liu**[2], **Jun Zhu**[1]†, **Qinsheng Zhang**[2]
[1]Department of Computer Science & Technology, Institute for AI, Tsinghua University
[2]NVIDIA

zkwthu@gmail.com; dcszj@mail.tsinghua.edu.cn;
{qinshengz,yongxinc,hanzim,mingyul}@nvidia.com

## Abstract

Masked diffusion models (MDMs) have emerged as a popular research topic for generative modeling of discrete data, thanks to their superior performance over other discrete diffusion models, and are rivaling the auto-regressive models (ARMs) for language modeling tasks. The recent effort in simplifying the masked diffusion framework further leads to alignment with continuous-space diffusion models and more principled training and sampling recipes. In this paper, however, we reveal that both training and sampling of MDMs are theoretically free from the time variable, arguably the key signature of diffusion models, and are instead equivalent to masked models. The connection on the sampling aspect is drawn by our proposed first-hitting sampler (FHS). Specifically, we show that the FHS is theoretically equivalent to MDMs' original generation process while significantly alleviating the time-consuming categorical sampling and achieving a $20\times$ speedup. In addition, our investigation raises doubts about whether MDMs can truly beat ARMs in text generation. We identify, for the first time, an underlying numerical issue, even with the commonly used 32-bit floating-point precision, which results in inaccurate categorical sampling. We show that it lowers the effective temperature both theoretically and empirically, and the resulting decrease in token diversity makes previous evaluations, which assess the generation quality solely through the incomplete generative perplexity metric, somewhat unfair.

## 1 Introduction

There are three primary paradigms of generative models. **Diffusion models** (Ho et al., 2020; Song et al., 2021c) have been the prevalent way for generative modeling of *continuous data* with both theoretical and empirical success. They are SOTA in image, speech, video synthesis (Dhariwal & Nichol, 2021; Karras et al., 2022; Chen et al., 2021; Ho et al., 2022) and serve as the cornerstone of large-scale text-to-image (Rombach et al., 2022; Balaji et al., 2022; Esser et al., 2024) and text-to-video (Gupta et al., 2023; Bao et al., 2024) generation systems. **Auto-regressive models** (ARMs) have dominated the generation of *discrete data* especially languages (Radford et al., 2018; 2019; Brown et al., 2020; Achiam et al., 2023; Touvron et al., 2023), due to the scalability and generalizability of the straightforward next-token-prediction mechanism based on transformer architec-

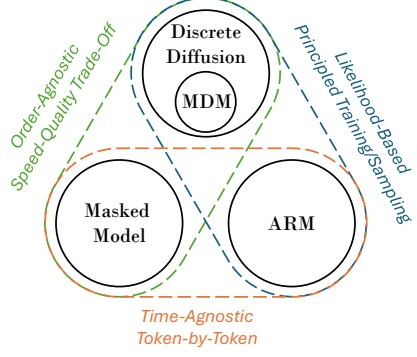

Figure 1: Trilemma of generative modeling for discrete data.

tures (Vaswani et al., 2017). **Masked models**, such as BERT (Devlin et al., 2019) for masked language modeling and MaskGIT (Chang et al., 2022) for masked image generation, are trained to

---

*Work done during an internship at NVIDIA; †The corresponding author.

reconstruct randomly masked tokens and sampled by order-agnostic decoding. They are an alternative approach to model *discrete data* while suffering from insufficient theoretical foundations.

Diffusion models have been extended to discrete data spaces with principled training and sampling (Austin et al., 2021; Campbell et al., 2022; Meng et al., 2022; Lou et al., 2023). Compared to ARMs, they predict all tokens simultaneously and offer a favorable trade-off between generation quality and sampling efficiency. Recently, **masked diffusion models** (MDMs), the leading variant of discrete diffusion formulations, are emerging as a promising contender of ARMs (Lou et al., 2023). Recent works (Shi et al., 2024; Sahoo et al., 2024) have simplified MDMs to align with the design space of diffusion models via continuous-time forward processes, training objectives, and sampling procedures, resulting in a unified view and empirical improvements. Positioned at the intersection of diffusion models and masked models, MDMs are considered promising as they inherit both the theoretical principles from diffusion models and the simple mechanism from masked models. Moreover, it is believed that MDMs can outperform ARMs in text generation when measured by the common generative perplexity metric (Lou et al., 2023; Shi et al., 2024; Sahoo et al., 2024).

However, we argue that the current understanding of MDMs is still quite limited in both theoretical and empirical aspects. In this paper[1], we conduct a thorough and comprehensive investigation and demonstrate that **MDMs are essentially a theoretically and empirically equivalent form of typical masked models while being complicated, inefficient, and numerically unstable**:

1. The **training objective** of MDMs is equivalent to that of masked models, differing only by nuanced likelihood-based loss weighting. The introduction of an additional time variable in the loss function provides little benefit in practice. (Section 3)

2. The **sampling process** of MDMs, being computationally expensive and inefficient, has a theoretically equivalent alternative (our first-hitting sampler) that is up to **20×** faster and mirrors the random-order, token-by-token decoding process of masked models. (Section 4)

3. The previously reported superiority of MDMs over ARMs on text generation stems from **numerical issues** that hack the generative perplexity metric during sampling by lowering the effective temperature, rather than genuine advantages. (Section 5)

Based on these findings, we argue that the community should reconsider investing efforts in MDMs. That being said, MDMs do provide theoretical insights and supplementary perspectives to masked models, but in practice, **the simpler masked models are not only sufficient but also free from above inefficiency and numerical instability issues**. Moreover, **scaling up MDMs on text encounters fundamental inference inefficiency challenges compared to ARMs**, as the bidirectional attention in masked models is incompatible with *KV caching*, a crucial technique for accelerating modern large language models (LLMs) that require long context length. This has been largely overlooked or intentionally downplayed by most existing works on diffusion language models[2]. Considering the critical role of infrastructure and cost-effective inference in the deployment of LLMs, MDMs (or masked models) lack a clear and compelling prospect to replace ARMs.

## 2 BACKGROUND: MASKED DIFFUSION MODELS (MDMs)

Let $\mathcal{X} = \{0, 1, \ldots, m-1\}$ be the discrete data space, with an extra mask token $m$ added to $\mathcal{X}$. Denote $\Delta^m = \{\boldsymbol{\pi} \in \mathbb{R}^{m+1} | \sum_{i=0}^{m} \pi_i = 1, \boldsymbol{\pi} \geq 0\}$ as the standard $m$-simplex. For any data token or mask token $x \in \mathcal{X}$, denote $\boldsymbol{e}_x \in \mathbb{R}^{m+1}$ as the corresponding one-hot vector. Continuous-time discrete-space masked diffusion models (MDMs) (Shi et al., 2024; Sahoo et al., 2024) can be defined akin to diffusion models, with a continuous-time forward noising process

$$q_{t|0}(x_t|x_0) = \text{Cat}(\alpha_t \boldsymbol{e}_{x_0} + (1 - \alpha_t)\boldsymbol{e}_m) \tag{1}$$

where $\alpha_t$ is the predefined *noise schedule* function satisfying $\alpha_0 \approx 1, \alpha_1 \approx 0$, and $\text{Cat}(\boldsymbol{\pi})$ denotes the *categorical distribution* over the class probabilities $\boldsymbol{\pi} \in \Delta^m$. The forward process has a time reversal for $s < t$ given $x_0$:

$$q_{s|t,0}(x_s|x_t, x_0) = \begin{cases} \text{Cat}(\boldsymbol{e}_{x_t}), & x_t \neq m \\ \text{Cat}\left(\frac{(1-\alpha_s)\boldsymbol{e}_m + (\alpha_s - \alpha_t)\boldsymbol{e}_{x_0}}{1 - \alpha_t}\right), & x_t = m \end{cases} \tag{2}$$

---

[1]We present our views in a straightforward manner. The earlier version is available in Appendix I.

[2]Given our findings, it may be more appropriate to call them **masked language models** if based on MDMs.

Following DDPM (Ho et al., 2020), the parameterized model is defined by replacing $\boldsymbol{e}_{x_0}$ in the reversal with a data prediction model $\boldsymbol{\mu}_\theta : \mathcal{X} \times \mathbb{R} \mapsto \Delta^m$:

$$p_\theta(x_s|x_t) := q(x_s|x_t, \boldsymbol{e}_{x_0} \leftarrow \boldsymbol{\mu}_\theta(x_t, t)) \tag{3}$$

and $\boldsymbol{\mu}_\theta$ is further parameterized by $\boldsymbol{f}_\theta : \mathcal{X} \times \mathbb{R} \mapsto \mathbb{R}^m$ as

$$\boldsymbol{\mu}_\theta(x_t, t) = \begin{cases} [\mathrm{softmax}(\boldsymbol{f}_\theta(x_t, t)), 0], & x_t = m \\ \boldsymbol{e}_{x_t}, & x_t \neq m \end{cases} \tag{4}$$

so that it satisfies (1) the predicted vector contains valid class probabilities sum to 1; (2) the predicted $x_0$ has zero probability of being the mask token; (3) if a token is already unmasked, it no longer changes. When $\alpha_0 \rightarrow 1, \alpha_1 \rightarrow 0$ and the number of timesteps tends to infinity, it is proven that the parameterized model $p_\theta$ has an *evidence lower bound* (ELBO) $\log p_\theta(x_0) \geq -\mathcal{L}_\infty$, where

$$\mathcal{L}_\infty = \int_0^1 \frac{\alpha_t'}{1 - \alpha_t} \mathbb{E}_{q_{t|0}(x_t|x_0)} \left[ \delta_{x_t, m} \boldsymbol{e}_{x_0}^\top \log \boldsymbol{\mu}_\theta(x_t, t) \right] \mathrm{d}t \tag{5}$$

is a time-weighted cross-entropy loss, $\alpha_t' = \frac{\mathrm{d}\alpha_t}{\mathrm{d}t}$, and $\delta_{x_t, m}$ is a indicator function. We refer to $\mathcal{L}_\infty$, the training objective, as the negative ELBO (NELBO).

**Multi-Dimensional Case**  For a token sequence $\boldsymbol{x} \in \mathcal{X}^L = \{0, 1, \ldots, m - 1, m\}^L$ of length $L$, MDMs choose a factorized forward process $q_{t|0}(\boldsymbol{x}_t|\boldsymbol{x}_0) = \prod_{l=1}^L q_{t|0}(x_t^{(l)}|x_0^{(l)})$ over different dimensions, where $x^{(l)}$ denotes the $l$-th token of $\boldsymbol{x}$. As a result, the reversal $q_{s|t,0}(\boldsymbol{x}_s|\boldsymbol{x}_t, \boldsymbol{x}_0) = \prod_{l=1}^L q_{s|t,0}(x_s^{(l)}|x_t^{(l)}, x_0^{(l)})$ and the parameterized model $p_\theta(\boldsymbol{x}_s|\boldsymbol{x}_t) = \prod_{l=1}^L q(x_s^{(l)}|x_t^{(l)}, \boldsymbol{e}_{x_0^{(l)}} \leftarrow \boldsymbol{\mu}_\theta^{(l)}(\boldsymbol{x}_t, t))$ also factorize. Here the network $\boldsymbol{\mu}_\theta : \mathcal{X}^L \times \mathbb{R} \mapsto (\Delta^m)^L$ predicts the probabilities at all positions at a time, and we use $\boldsymbol{\mu}_\theta^{(l)}$ to denote the $l$-th column of $\boldsymbol{\mu}_\theta$. The ELBO loss in Eqn. (5) under multi-dimension can be written as

$$\mathcal{L}_\infty^{(L)} = \int_0^1 \frac{\alpha_t'}{1 - \alpha_t} \mathbb{E}_{q_{t|0}(\boldsymbol{x}_t|\boldsymbol{x}_0)} \left[ \sum_{l:x_t^{(l)}=m} \boldsymbol{e}_{x_0^{(l)}}^\top \log \boldsymbol{\mu}_\theta^{(l)}(\boldsymbol{x}_t, t) \right] \mathrm{d}t \tag{6}$$

**Context of Discrete Diffusion Models**  MDMs described above are a simplified version of the best-performing masked (or absorbing) case in discrete-space diffusion models. Discrete diffusion models, originated from D3PM (Austin et al., 2021), rely on discrete-time or continuous-time Markov chains to model transitions in discrete space. Notably, *concrete score* (Meng et al., 2022) in discrete diffusion acts as an analog of the score function in continuous diffusion, and a recent work SEDD (Lou et al., 2023) proposes *score entropy* for robust and scalable learning of the concrete score. The model definition (Markov chain, score parameterization), training objective (diffusion-weighted denoising score entropy) and sampling procedure (Tweedie $\tau$-leaping) of SEDD Absorb can be proven equivalent to the simplified expressions (Eqn. (1) (3) (4) (5)) in MDMs. Interested readers can refer to Appendix D for further details.

## 3 REVISITING THE TRAINING OF MDMS

MDMs are defined and trained by the continuous-time forward process (Eqn. (1)), time-dependent network parameterization (Eqn. (4)) and continuous-time ELBO (Eqn. (5)). However, different from continuous-time diffusion models (Song et al., 2021c), the evolution of $\boldsymbol{x}_t$ is discrete. The evolution trajectories of $(\boldsymbol{x}_t, t)$ are like pairs of "phenotype" and "genotype", where the continuous changes in time $t$ may not be reflected on the observable traits of $\boldsymbol{x}_t$. In this section, we aim to disentangle the internal time variable $t$ and the external traits of the masked sequence $\boldsymbol{x}_t$ in the training of MDMs.

### 3.1 REFORMULATING THE ELBO WITH THE NUMBER OF MASKED TOKENS

Previous works (Shi et al., 2024; Sahoo et al., 2024) show the invariance of the ELBO to the noise schedule $\alpha_t$ by performing the time change-of-variable $\gamma = \log(1 - \alpha_t)$ or $\lambda = \log \frac{\alpha_t}{1-\alpha_t}$ following VDM (Kingma et al., 2021). However, this does not get to the essence as they still rely on an internal continuous time. In the following proposition, we show that the sequence NELBO of MDMs can be expressed as a partition by the number of masked tokens instead of the continuous time.

**Proposition 3.1** (ELBO by the Number of Masked Tokens). *For $\boldsymbol{x}_0$ with sequence length $L$, denote $\boldsymbol{x}_n$ as a sequence with $n$ masked tokens, and $\tilde{q}(\boldsymbol{x}_n|\boldsymbol{x}_0)$ as the discrete forward process which randomly and uniformly masks $n$ tokens of $\boldsymbol{x}_0$. Suppose the noise schedule $\alpha_t$ satisfies $\alpha_0 = 1, \alpha_1 = 0$. The sequence NELBO in Eqn.* (6) *can be reformulated as*

$$\mathcal{L}_\infty^{(L)} = -\sum_{n=1}^{L} \mathbb{E}_{\tilde{q}_{n|0}(\boldsymbol{x}_n|\boldsymbol{x}_0)} \left[ \frac{1}{n} \sum_{l:x_n^{(l)}=m} \boldsymbol{e}_{x_0^{(l)}}^\top \log \bar{\boldsymbol{\mu}}_\theta^{(l)}(\boldsymbol{x}_n) \right] \tag{7}$$

*where*

$$\log \bar{\boldsymbol{\mu}}_\theta(\boldsymbol{x}_n) = \mathbb{E}_{\alpha_n \sim \mathcal{B}(L-n+1,n)} \left[ \log \boldsymbol{\mu}_\theta(\boldsymbol{x}_n, \alpha^{-1}(\alpha_n)) \right], \tag{8}$$

$\alpha^{-1}$ *is the inverse function of $\alpha_t$ satisfying $\alpha^{-1}(\alpha_t) = t$, and $\mathcal{B}(a,b)$ denotes the Beta distribution with shape parameters $a, b > 0$.*

This expression offers two aspects of theoretical insights:

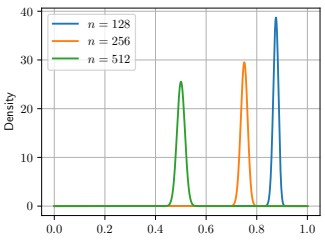

Figure 2: Probability density function (PDF) of $\mathcal{B}(L-n+1,n)$ with $L = 1024$.

**Mixture of Experts** From Eqn. (8), the time-dependent network $\boldsymbol{\mu}_\theta(\boldsymbol{x}, t)$ implicitly parameterizes a time-independent network $\bar{\boldsymbol{\mu}}_\theta(\boldsymbol{x})$ by aggregating the logarithm at the same $\boldsymbol{x}$ but different $t$, which can be seen as an ensemble. The time $t$ is sampled unevenly so that $\alpha_t$ follows a Beta distribution $\mathcal{B}(L-n+1, n)$. This distribution has the mode (peak) $\frac{L-n}{L-1}$ and variance $\frac{n(L-n+1)}{(L+1)^2(L+2)} \leq \frac{1}{4(L+2)}$. With a large sequence length $L$, the variance is small and the distribution is concentrated around the mode, as illustrated in Figure 2. Moreover, under the best-performing linear schedule $\alpha_t = 1 - t$ in MDMs (Lou et al., 2023; Shi et al., 2024; Sahoo et al., 2024), the mode of $t$ is $\frac{n-1}{L-1}$, close to the *masked ratio* $\frac{n}{L}$. Therefore, the time variable $t$ can be seen as a continuous relaxation and smoothing of the masked ratio, and we can directly condition the network on the discretely distributed masked ratio instead of the continuous time while yielding similar performance (Appendix J.1).

**Discrete ELBO** From Eqn. (7), the sequence NELBO can be expressed discretely with the time-agnostic network $\bar{\boldsymbol{\mu}}_\theta(\boldsymbol{x})$. Therefore, Eqn. (7) can serve as a NELBO of masked models in a straightforward way: *uniformly* choose the number of masked tokens $n$ from $\{1, \ldots, L\}$, *uniformly* mask $n$ random tokens in $\boldsymbol{x}_0$ to obtain $\boldsymbol{x}_n$, and compute the *average* cross-entropy loss of $\bar{\boldsymbol{\mu}}_\theta(\boldsymbol{x})$ on these $n$ positions. The weighting $\frac{1}{n}$ in this NELBO resembles the *likelihood weighting* in diffusion models (Song et al., 2021b; Kingma et al., 2021; Lu et al., 2022a; Zheng et al., 2023b), facilitating *maximum likelihood training* of masked models. Note that early works on order-agnostic autoregressive models (Uria et al., 2014; Hoogeboom et al., 2021a) already reveal this weighting from a different perspective[3]. While in the context of masked models, there are few discussions on the ELBO. Discussions on related work are placed in Appendix B.

## 3.2 TIME-INDEPENDENT NETWORK PARAMETERIZATION

When the original network $\boldsymbol{\mu}_\theta$ is parameterized without the time input, we have $\bar{\boldsymbol{\mu}}_\theta = \boldsymbol{\mu}_\theta$ in Eqn (7). In this case, the training of MDMs is completely free from the time variable and behaves like masked models. The rationality of time-independent network parameterization has been discussed in recent works (Ou et al., 2024; Sahoo et al., 2024). Here we restate this conclusion with our simplified notations from the perspective of the optimal model.

**Proposition 3.2** (Optimal Masked Diffusion Model). *Given unlimited model capacity, the optimal network $\theta^*$ that minimizes the NELBO in Eqn.* (6) *satisfies*

$$\boldsymbol{\mu}_{\theta^*}^{(l)}(\boldsymbol{x}, t) = \mathbb{E}_{\tilde{q}_{0|N(\boldsymbol{x})}(\boldsymbol{x}_0|\boldsymbol{x})} \left[ \boldsymbol{e}_{x_0^{(l)}} \right] \tag{9}$$

*where $N(\boldsymbol{x})$ is a deterministic function that counts the number of masked tokens in $\boldsymbol{x}$, and $\tilde{q}_{0|n}(\boldsymbol{x}_0|\boldsymbol{x}_n)$ is the posterior distribution of the discrete forward process $\tilde{q}_{n|0}(\boldsymbol{x}_n|\boldsymbol{x}_0)$.*

---

[3]The relation between ELBOs of order-agnostic ARMs and MDMs was also mentioned in a recent work (Ou et al., 2024), while they only consider an originally time-agnostic network instead of mixture-of-experts.

From the above expression, the optimal MDM is irrelevant to the time variable, justifying the feasibility of removing the time input. Besides, it can be extended to a general weighted cross-entropy loss $\mathcal{L}_{\boldsymbol{w}}^{(L)} = -\sum_{n=1}^{L} w_n \mathbb{E}_{\tilde{q}_{n|0}(\boldsymbol{x}_n|\boldsymbol{x}_0)} \left[ \sum_{l:x_n^{(l)}=m} \boldsymbol{e}_{x_0^{(l)}}^{\top} \log \boldsymbol{\mu}_{\theta}^{(l)}(\boldsymbol{x}_n) \right]$ of masked models. $\mathcal{L}_{\boldsymbol{w}}^{(L)}$ with arbitrary positive weights $\boldsymbol{w} > 0$ yields the same optimal solution as Eqn. (9), thus acting as a surrogate objective of the NELBO. This theoretically supports a wide range of objectives for training masked models, such as the loss in MaskGIT (Chang et al., 2022).

## 3.3 PRACTICAL CONSIDERATIONS

While there are theoretically equivalent variants for training MDMs (continuous-time/discrete ELBO, time-conditioned/time-independent network), these choices may have practical implications due to differences in network inputs and loss variances. We present some training comparisons and our attempts to improve training (e.g., variance reduction, flow matching) in Appendix J.1. Overall, all options yield similar performance, and the low-discrepancy sampler (Kingma et al., 2021), when applied to time or the number of masked tokens, can significantly reduce the loss variance.

Note that while several works (Lou et al., 2023; Shi et al., 2024) suggest that MDMs are competitive with ARMs in language modeling (beating GPT-2 (Radford et al., 2019) when measured by test/zero-shot perplexity), a more fair comparison (retraining ARMs with the same configurations, Appendix J.1) (Sahoo et al., 2024) indicates that MDMs are only advantageous in language understanding tasks (surpassing ARMs and BERT on the GLUE metric (Wang et al., 2018)).

## 4 REVISITING THE SAMPLING OF MDMS

In the previous section, we demonstrate how the training of MDMs, both theoretically and empirically, can be disentangled with the continuous time variable and behave like masked models. In this section, we turn our attention to the sampling of MDMs, which is also performed in continuous time and seems distinct from masked models. We aim to address its current inefficiency problem as well as establish essential insights into its connection with masked models.

### 4.1 INEFFICIENCY OF CURRENT SAMPLING

MDMs are sampled in an ancestral way following the parameterized reverse-time process in Eqn. (3). Specifically, the sampling step $\boldsymbol{x}_t \to \boldsymbol{x}_s$ from time $t$ to $s < t$ can be expressed as

$$x_s^{(l)} \begin{cases} = x_t^{(l)}, & x_t^{(l)} \neq m \\ \sim \text{Cat}\left( \frac{(1-\alpha_s)\boldsymbol{e}_m + (\alpha_s - \alpha_t)\boldsymbol{\mu}_{\theta}^{(l)}(\boldsymbol{x}_t, t)}{1 - \alpha_t} \right), & x_t^{(l)} = m \end{cases}, \quad \text{for every } l \qquad (10)$$

Given the number of sampling steps $N$, the sampling process involves first discretizing the timesteps as $0 = t_0 < t_1 < \cdots < t_N = 1$, and then performing reverse steps $t_N \to t_{N-1} \to \cdots \to t_0$ according to Eqn. (10). Notable characteristics of MDM's sampling include: (1) Any mask token can only be unmasked once with no further changes. (2) Each sampling step requires a forward pass through the network $\boldsymbol{\mu}_{\theta}$ and conducting at most $L$ times of $|\mathcal{X}|$-dimensional categorical sampling, where $L$ is the sequence length and $|\mathcal{X}|$ is the vocabulary size. (3) The number of sampling steps $N$ can be significantly larger than $L$, and a single sampling step may result in no changes to any token in the sequence. (4) As MDMs are trained with the continuous-time ELBO which assumes an infinite number of reverse steps, it is theoretically rigorous to employ an equivalently large $N$.

Recent works propose a simple *caching strategy* (Ou et al., 2024; Sahoo et al., 2024) to speedup the sampling of MDMs: when the network $\boldsymbol{\mu}_{\theta}$ is parameterized without time input[4], and the sequence is not changed in a sampling step $t \to s$ (i.e., $\boldsymbol{x}_s = \boldsymbol{x}_t$), we can reuse the network output at the last step as $\boldsymbol{\mu}_{\theta}(\boldsymbol{x}_s) = \boldsymbol{\mu}_{\theta}(\boldsymbol{x}_t)$. As the sequence changes at most $L$ times during sampling, the number of function evaluations (NFE) can be reduced to no more than $L$. However, sampling with the caching strategy still suffers from two major inefficiency problems:

---

[4]In our practice, the time-dependent network also exhibits no performance degradation with the caching strategy, so this assumption is unnecessary.

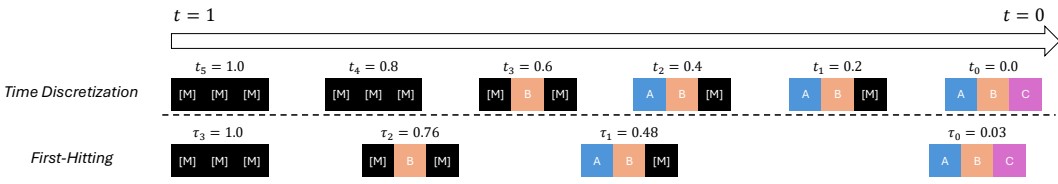

Figure 4: Illustration of the first-hitting sampler in comparison to the original sampling procedure.

**Categorical Sampling is Time-Consuming**  In diffusion models, NFE is an efficient indicator of the sampling speed, as the computation overhead beyond the network forward passes is negligible. However, in MDMs, the Gumbel-based[5] categorical sampling, which requires sampling a total number of $\mathcal{O}(NL|\mathcal{X}|)$ uniform variables and performing logarithmic operations on them, can be expensive compared to network evaluations. As illustrated in Figure 3a, when the number of sampling steps $N \gg L$, the sampling time scales with $N$ instead of the NFE. Categorical sampling steps that do not result in token changes are wasted, as they contribute no information gain.

**Caching Strategy Degrades in Batched Sampling**  When using the caching strategy in batched sampling, the network output can only be reused directly when all the sequences in the batch remain unchanged after a sampling step[6]. Suppose the batch size is $B$, and the default linear noise schedule $\alpha_t = 1 - t$ as well as uniform timesteps $t_k = \frac{k}{N}$ is used. The *expected NFE* under the caching strategy can be derived as $N(1 - (1 - \frac{1}{N})^{BL})$ (proof in Appendix E), similar to the $B = 1$ case in Ou et al. (2024). As $\lim_{N\to\infty} N(1 - (1 - \frac{1}{N})^{BL}) = BL$, the NFE is no longer upper bounded by the sequence length but scales with the batch size (Figure 3b).

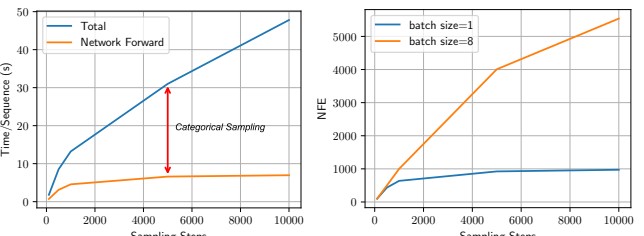

(a) Sampling time per sequence (caching strategy, batch size=1)

(b) NFE (caching strategy)

Figure 3: Illustration of sampling ineffciency using pretrained models of MDLM (Sahoo et al., 2024) ($L = 1024$).

### 4.2 FIRST-HITTING SAMPLER

The current sampling methods of MDMs, including the caching strategy, are neither efficient nor insightful into the essence of MDMs. To address this, we reexamine the sampling step in Eqn. (10).

When the number of sampling steps $N \to \infty$ and the maximum step size $\max_{1 \le i \le N} |t_i - t_{i-1}| \to 0$, Eqn. (10) tends to an infinitesimal jump. In this case, the reverse sampling process becomes a continuous-time Markov chain

---

**Algorithm 1** First-Hitting Sampling of MDMs

**Require:** the sequence length $L$, the vocabulary $\mathcal{X} = \{0, \dots, m - 1, m\}$ where $m$ is the mask token, the noise schedule $\alpha_t$ and its inverse function $\alpha^{-1}$, the pretrained masked diffusion model $\boldsymbol{\mu}_\theta$
1: $\boldsymbol{x}_L \leftarrow [m \ m \ \dots \ m]$
2: $\tau_L \leftarrow 1$
3: **for** $n \leftarrow L$ **to** 1 **do**
4:     Sample $u_n \sim \mathcal{U}(0, 1)$
5:     $\tau_{n-1} \leftarrow \alpha^{-1}(1 - u_n^{1/n}(1 - \alpha_{\tau_n}))$
6:     $\boldsymbol{\mu}_n \leftarrow \boldsymbol{\mu}_\theta(\boldsymbol{x}_n, \tau_{n-1})$
7:     Randomly and uniformly select an index $l$ from $\{i : x_n^{(i)} = m\}$ (i.e., masked positions in $\boldsymbol{x}_n$)
8:     $\boldsymbol{x}_{n-1} \leftarrow \boldsymbol{x}_n, x_{n-1}^{(l)} \leftarrow x \sim \text{Cat}(\boldsymbol{\mu}_n^{(l)})$
9: **end for**
**Output:** $\boldsymbol{x}_0$

---

(or Markov process), where each mask token is unmasked at some moment according to the network prediction. Our *key insight* involves three folds: (1) *Whether a mask token will transit or not* during a time interval $[s, t]$ is independent of the network. The network output only determines *which token is the transition target* given the condition that the transition happens. (2) The transition probability $\frac{\alpha_s - \alpha_t}{1 - \alpha_t}$ is equal for masked tokens at different positions. Therefore, each mask token has the same probability of being first unmasked. (3) The *first-hitting time*, which denotes the first moment any of the remaining masked tokens is unmasked, can be *analytically sampled*:

**Proposition 4.1** (Analytic Sampling of First-Hitting Time)**.** *Denote $\tau_L = 1$ as the initial time. Suppose there are $n$ masked tokens, and the last time a token is unmasked happens at $\tau_n$, then the*

---

[5]We will introduce Gumbel-based categorical sampling in the next section.

[6]We can reuse only the unchanged part of a batch, but this potentially reduce parallel efficiency.

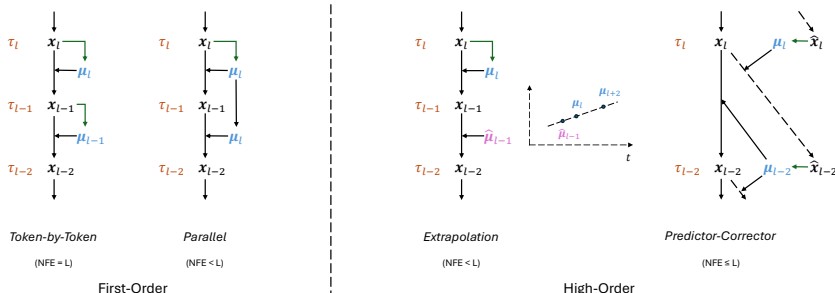

Figure 5: Variants of the first-hitting sampler. $\boldsymbol{x}_l$ denotes the sequence with $l$ remaining mask tokens, and $\boldsymbol{\mu}_l = \boldsymbol{\mu}_\theta(\boldsymbol{x}_l, \tau_{l-1})$ denotes the network prediction at the step $l$.

*next time a token is unmasked can be analytically sampled by*

$$\tau_{n-1} = \alpha^{-1}(1 - u_n^{1/n}(1 - \alpha_{\tau_n})), \quad u_n \sim \mathcal{U}(0,1) \tag{11}$$

*where $\mathcal{U}(0,1)$ is the uniform distribution on $[0,1]$.*

As outlined in Algorithm 1, by recursively sampling the next time when any of the remaining mask tokens is first unmasked, then uniformly choosing a mask token and unmasking it according to the network output, we obtain a token-by-token sampling procedure of MDMs. Denote $\boldsymbol{x}_n$ as the sequence with $n$ remaining mask tokens. Since the transition $\boldsymbol{x}_n \to \boldsymbol{x}_{n-1}$ can be considered to happen in the infinitesimal step $\tau_{n-1} + \mathrm{d}t \to \tau_{n-1}$, using the network output $\boldsymbol{\mu}_\theta(\boldsymbol{x}_n, \tau_{n-1})$ at time $\tau_{n-1}$ incurs no approximation errors. Therefore, the first-hitting sampler (FHS) is *theoretically equivalent* as simulating the continuous-time reverse Markov sampling process. We illustrate the comparison between the FHS and the original sampling procedure in Figure 4.

The FHS demonstrates appealing properties:

**Tackling the Sampling Inefficiency**   The FHS can tackle the two inefficiency problems described in Section 4.1. Firstly, as the categorical sampling is only conducted for determining the transition target of the single chosen mask token at each step, the total computation cost is reduced to $\mathcal{O}(L|\mathcal{X}|)$. Secondly, the first-hitting time $\tau_n$ can be sampled independently and asynchronously across different samples in a batch, avoiding performance degradation in batched sampling.

**Connection to the Sampling of Masked Models**   When the network parameterization is independent of the time, the FHS in Algorithm 1 can be completely free from the time and become a token-by-token decoding process akin to masked models. This connection serves as supporting evidence for the typical sampling procedure of masked models, as it is theoretically equivalent to the more principled reverse Markov sampling process of MDMs.

### 4.3 PARALLEL DECODING AND HIGH-ORDER VARIANTS

The token-by-token decoding process of MDMs can be extended to parallel decoding by unmasking multiple tokens per step, as the network $\boldsymbol{\mu}_\theta$ predicts tokens at all positions. This enables speed-quality trade-offs similar to diffusion models. As illustrated in Figure 5, parallel decoding essentially reuses the previous network output to reduce the NFE, thus functioning as an approximation method.

To reduce the approximation error, we follow the recipes of high-order diffusion solvers (Karras et al., 2022; Zhang & Chen, 2022; Lu et al., 2022b; Zheng et al., 2023a) to develop high-order samplers of MDMs. We propose two variants: one based on extrapolating previous network outputs, and the other utilizing a predictor-corrector method to refine the samples (algorithms in Appendix G.1).

## 5 ARE MDMS BETTER THAN ARMS? A CRITICAL FAULT IN LOW-PRECISION GUMBEL-BASED CATEGORICAL SAMPLING

Before we proceed to verify the effectiveness of our proposed first-hitting sampler, we have to point out a critical fault in MDMs' original sampling implementation. As suggested by previous

> There is the following definition:
> The "right lane" on the lane lane lane. From the lane lane lane from lane lane to lane lane on a lane in lane lane on a front lane.
> From lane lane lane the lane lane lane on the right lane from lane front lane lane to top lane.
> From the right lane from a lane lane lane with the lane on the lane lane lane. The "that lane lane" on the rear lane.

Figure 6: Segment of generated text by SEDD Absorb (Lou et al., 2023) at 50k sampling steps.

Table 1: Maximum Gumbel under different floating-point precisions.

| Data Type | Sign | Structure (bits) Exponent | Fraction | Maximum Value ($< 1$) Representable | Maximum Gumbel |
|---|---|---|---|---|---|
| float32 | 1 | 8 | 23 | $1 - 2^{-24} \approx 0.9999999404$ | $-\log(-\log(1 - 2^{-24})) \approx 16.6355$ |
| float64 | 1 | 11 | 52 | $1 - 2^{-53} \approx 0.999999999999999889$ | $-\log(-\log(1 - 2^{-53})) \approx 36.7368$ |

works (Lou et al., 2023; Shi et al., 2024; Sahoo et al., 2024), MDMs seem to surpass ARMs with a sufficient number of sampling steps when measured by the generative perplexity (Gen PPL)[7], as shown in Figure 7a. However, in this section, we identify for the first time a hidden numerical issue existing in previous codebases that makes this observation questionable.

## 5.1 LOW TOKEN DIVERSITY UNDER NUMEROUS SAMPLING STEPS

Empirically, a reduction in Gen PPL is observed by increasing the inference budget. In particular, an exceptionally low Gen PPL ($< 15$) is achieved when the number of sampling steps approaches 50k.

However, when we check the generated content, we discover that the quality is compromised by low token diversity (an extreme case is shown in Figure 6). We further quantify this phenomenon by measuring the sentence entropy (Figure 7b). With the original sampler, the Gen PPL of MDMs surpasses ARMs at around 2k steps, but *the entropy is always lower and keeps decreasing*.

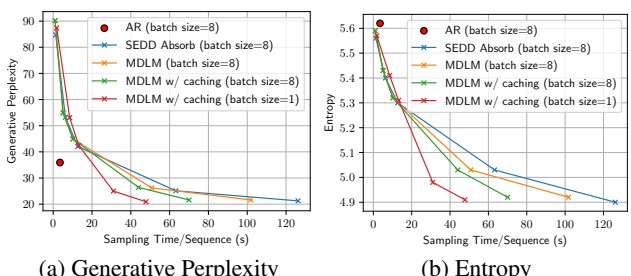

(a) Generative Perplexity     (b) Entropy

Figure 7: Comparisons of different models (AR, SEDD Absorb (Lou et al., 2023), MDLM (Sahoo et al., 2024)) trained on OpenWebText (Gokaslan et al., 2019) with the same network architecture and configurations. We generate 64 samples using their original codebase on a single NVIDIA RTX A6000 GPU, and vary the sampling steps $N \in \{100, 500, 1000, 5000, 10000\}$.

This low generation quality is unexpected, as theory suggests that increasing sampling steps should yield lower discretization errors and more faithfully reflect the true model performance. We therefore consider this a hidden implementation issue and investigate further to identify the root cause.

## 5.2 IDENTIFYING THE NUMERICAL PRECISION PROBLEM

Our *key observation* is that, when we alter the floating-point precision during sampling from 32-bit to 64-bit, the entropy returns to a normal level similar to ARMs, but with a generative perplexity $\approx 100$. After careful ablations, we identify the root cause as the inaccuracy in previous Gumbel-based categorical sampling. To sample from a categorical distribution with class probabilities $\{\pi_i\}_{i=1}^K$, Gumbel-max trick[8] is used by first sampling $K$ independent uniform variables $u_i \sim \mathcal{U}(0, 1)$, then transforming them into samples from the standard Gumbel distribution $\mathcal{G}(0, 1)$ by $g_i = -\log(-\log u_i)$, and finally obtaining the categorical sample $n = \arg\max_i(\log \pi_i + g_i)$. The operation $g = -\log(-\log u)$ theoretically maps $u \in [0, 1]$ to $g \in (-\infty, +\infty)$. But due to the limited representation ability of floating-point numbers in implementation, $u$ is constrained to $[0, 1 - \epsilon]$ and $g$ is constrained to $(-\infty, M]$, as shown in Table 1. Therefore, the sample $g$ instead follows a *truncated Gumbel distribution*, denoted $\mathcal{TG}(0, 1, M)$, which refers to the Gumbel distri-

---

[7]The evaluation metrics used in this paper are introduced in Appendix H.1.

[8]A brief introduction to Gumbel tricks is provided in Appendix F

```python
def sample_categorical_Gumbel(probs):

    u = torch.rand(                          torch.rand(...dtype=torch.float32)                    32-bit
        *probs.shape,
        device=probs.device,                torch.rand(...dtype=torch.float64)                    64-bit
        dtype=...
        ...                                  torch.rand(...dtype=torch.float64) * 0.9999999404     64-bit + truncation

    return (probs / (-u.log())).argmax(dim=-1)
```

Figure 8: Code for different versions of Gumbel-based categorical sampling. The operation $\text{argmax}_i(\log \pi_i - \log(-\log u_i))$ is simplified to $\text{argmax}_i(\pi_i/(-\log u_i))$ to save computation cost.

bution $\mathcal{G}(0,1)$ conditioned on $g \leq M$. This tricky difference theoretically makes the categorical sampling inaccurate, i.e., $\text{argmax}_i(\log \pi_i + g_i)$ no longer follows the class probabilities $\{\pi_i\}_{i=1}^K$.

To verify that truncation is the fundamental issue, we conduct ablations by only modifying the categorical sampling code. As shown in Figure 8, we manually scale 64-bit uniform samples to match the truncation in the 32-bit case. We then randomly generate 8 samples with 2048 steps and compare the average generative perplexity and entropy in Table 2. The similar results between the 32-bit and truncated 64-bit cases confirm the impact of truncation.

Table 2: Results with different versions of categorical sampling.

| Version | Gen PPL | Entropy |
|---|---|---|
| 32-bit | 31.24 | 5.17 |
| 64-bit | 126.11 | 5.66 |
| 64-bit + trunc | 28.64 | 5.12 |

*Remark* 5.1. Note that auto-regressive LLMs like Llama (Touvron et al., 2023) and Mistral (Jiang et al., 2023) use `torch.multinomial` for categorical sampling, which is also implemented with the Gumbel-max trick in the low-level C++ code of PyTorch. In contrast, we find **the token-by-token decoding process of ARMs and MDMs (by our first-hitting sampler) does not suffer from notable numerical issues under 32-bit precision** (illustrations and explanations in Appendix J.2.2).

## 5.3 CATEGORICAL SAMPLING WITH TRUNCATED GUMBEL

In the previous section, we empirically observe that truncated Gumbel-based categorical sampling reduces token diversity. Surprisingly, such effects can be precisely depicted in *closed-form*.

**Proposition 5.2** (Closed-Form Categorical Sampling with Truncated Gumbel). *Suppose the class probabilities are sorted as $\pi_1 \leq \cdots \leq \pi_K$, and $g_i \sim \mathcal{TG}(0, 1, M)$ are truncated Gumbel samples with maximum value $M$. Denote $\pi_0 = 0$. For $1 \leq n \leq K$, we have $P(\text{argmax}_i(\log \pi_i + g_i) = n) = \pi_n \sum_{i=1}^n \beta(i)$, where*

$$\beta(i) = \frac{e^{\left(K+1-i-\frac{\sum_{k=i}^K \pi_k}{\pi_i}\right)e^{-M}} - e^{\left(K+1-i-\frac{\sum_{k=i}^K \pi_k}{\pi_{i-1}}\right)e^{-M}}}{\sum_{k=i}^K \pi_k} \geq 0 \qquad (12)$$

*To the best of our knowledge, this formulation has not been revealed in previous works.* Intuitively, with truncated Gumbel, the original class probabilities $\pi_n$ are shifted to $\pi'_n = \pi_n \sum_{i=1}^n \beta(i)$. This has two main implications: (1) As $\beta(i) \geq 0$ and $\pi_n$ are sorted, if $\pi_{n_1} > \pi_{n_2}$, the adjusted class probabilities satisfy $\frac{\pi'_{n_1}}{\pi'_{n_2}} > \frac{\pi_{n_1}}{\pi_{n_2}}$. This indicates that relatively larger probabilities are further amplified, creating an effect similar to lowering the temperature. (2) In the sampling step, the probability of unmasking is adjusted based on the network output, resulting in unequal unmasking probabilities at different positions in a sequence. This implies that some tokens are prioritized to be unmasked, further reducing the randomness and overall entropy.

In both aspects, the inaccurate categorical sampling deviates from theoretical correctness and reduces the generation diversity, leading to unfair evaluations of MDMs' generative performance.

## 6 A FAIR EVALUATION OF MDMS' GENERATION

In this section, we will fairly evaluate the generation performance of MDMs and examine the impact of our proposed sampler and the temperature. Our experiments are based on the codebase of MDLM (Sahoo et al., 2024) which is inherited from SEDD (Lou et al., 2023). We fix the categorical sampling to 64-bit floating-point precision so that the numerical truncation is negligible. We

directly use pretrained models (AR, SEDD Absorb, MDLM) provided by MDLM, which share the same network architecture and were trained with the same configuration. Additional experiment details are provided in Appendix H. We display some generated text in Appendix J.3.1 to illustrate the token diversity under different sampling strategies.

## 6.1 ORIGINAL SAMPLER V.S. FIRST-HITTING SAMPLER

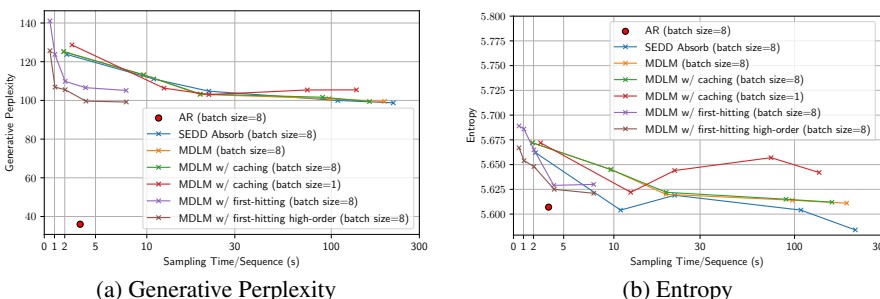

(a) Generative Perplexity          (b) Entropy

Figure 9: Comparisons of different models after fixing the categorical sampling to 64-bit. We additional compare our propose first-hitting sampler with steps $N \in \{64, 128, 256, 512, 1024\}$.

Figure 9 compares both the generative perplexity and the entropy of different models. For the baselines, SEDD is sampled by their analytic sampler (Tweedie $\tau$-leaping), and MDLM is sampled with and without the caching strategy. For our first-hitting sampler, the parallel decoding is performed by unmasking the same number of tokens per step. High-order variants employ the extrapolation strategy when the number of sampling steps $N \leq 128$, and the predictor-corrector strategy otherwise.

After the numerical problem is fixed, the entropy returns to a normal level (5.60~5.70) for all models. Besides, our sampler can be up to 20× faster than previous sampling strategies of MDMs in terms of the wall-clock time[9]. Despite the notable speedup, the true generative perplexity of MDMs is revealed to be around 100, significantly lagging behind that of counterpart ARMs ($< 40$).

## 6.2 TRADING OFF GENERATIVE PERPLEXITY AND ENTROPY VIA TEMPERATURE

The truncation effect of 32-bit floating-point numbers creates a trade-off between generative perplexity and entropy by varying the number of sampling steps (Figure 7). This trade-off arises from a tricky interplay of inaccurate categorical sampling and the approximation error at limited discretization steps. In Figure 10, we demonstrate that this trade-off can be achieved at a lower time cost by using the correct sampling (our 1024-step high-order sampler) and manually adjusting the temperature within the range $[0.8, 1.0]$. The trade-off curve of our method is slightly better than the original MDM sampling, while still significantly lagging behind ARMs.

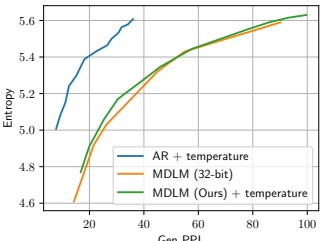

Figure 10: Trade-off of generative perplexity and entropy.

## 7 CONCLUSION

In this work, we advance our understanding of masked diffusion models (MDMs) by revealing their theoretical equivalence to masked models and uncovering a hidden numerical issue that compromised the fairness of previous evaluations of MDMs' generative performance. Our findings challenge earlier claims that MDMs can surpass ARMs in text generation. Despite these negative results, we acknowledge that our text-based experiments may inherently favor ARMs, as text naturally follows a left-to-right order that ARMs are better suited to model. Nevertheless, we believe that MDMs may hold potential for applications where an order-agnostic data structure is a key prior, and in practice, simply using masked models may be a better choice.

---

[9]The efficiency gains (measured by inference wall-clock time) can depend on many factors and may not be as large as 20x in other settings (analyzed in Appendix J.3).

## ACKNOWLEDGMENTS

The team would like to thank Aaron Lou, Cheng Lu from OpenAI, and Jiaxin Shi from Google DeepMind for their valuable discussions and comments. K. Z and J. Z were also supported by the National Natural Science Foundation of China (Nos. 62350080, 62106120, 92270001), Tsinghua Institute for Guo Qiang, and the High Performance Computing Center, Tsinghua University; J. Z was also supported by the XPlorer Prize.

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

CONTENTS

## A NOTATIONS AND DEFINITIONS

**Numbers and Arrays**

| | |
|---|---|
| $x$ | A scalar representing a discrete token |
| $\boldsymbol{x}$ | A vector representing a sequence of discrete tokens |
| $x^{(l)}$ | The $l$-th element of $\boldsymbol{x}$ |
| $x_t, \boldsymbol{x}_t$ | The state(s) at time $t$ |
| $\boldsymbol{x}_n$ | The sequence with $n$ masked tokens |
| $t$ | The continuous time |
| $m$ | The mask token |
| $n$ | The number of masked tokens in a sequence |
| $\boldsymbol{\mu}$ | A matrix, where the $l$-th column represents the predicted transition probabilities at the $l$-th position in a sequence |
| $\boldsymbol{\mu}^{(l)}$ | The $l$-th column of $\boldsymbol{\mu}$ |
| $\boldsymbol{\pi}$ | The class probabilities |
| $\pi_i$ | The $i$-th element of $\boldsymbol{\pi}$ |
| $L$ | The sequence length |
| $N$ | The number of sampling steps |
| $B$ | The batch size |
| $\theta$ | The neural network parameters |
| $\tau$ | The first-hitting time |
| $\mathcal{L}_\infty$ | The continuous-time NELBO loss for a single token |
| $\mathcal{L}_\infty^{(L)}$ | The continuous-time NELBO loss for a sequence of length $L$ |

**Sets**

| | |
|---|---|
| $\mathbb{R}$ | The set of real numbers |
| $\mathcal{X}$ | The discrete data space (vocabulary) $\{0, 1, \ldots, m\}$ where $m$ is the added mask token |
| $\Delta^m$ | The standard $m$-simplex $\left\{ \boldsymbol{\pi} \in \mathbb{R}^{m+1} \mid \sum_{i=0}^m \pi_i = 1, \boldsymbol{\pi} \geq 0 \right\}$ |

**Functions**

| | |
|---|---|
| $\alpha_t$ | The pre-defined noise schedule, which is a decreasing function of time $t$ |
| $\alpha_t'$ | The derivative of the noise schedule w.r.t. the time |
| $\alpha^{-1}(a)$ | The inverse function of the noise schedule satisfying $\alpha_{\alpha^{-1}(a)} = a$ |
| $\delta_{x,y}$ | The indicator function (1 when $x = y$ and 0 when $x \neq y$) |
| $\boldsymbol{e}_x$ | The one-hot vector of the token $x$ |
| $\boldsymbol{\mu}_\theta(\boldsymbol{x}, t)$ | The network prediction given the sequence $\boldsymbol{x}$ and the time $t$ as input |
| $\mathrm{softmax}(\boldsymbol{z})$ | The Softmax operation to transform logits into class probabilities |
| $\log \boldsymbol{\mu}$ | The element-wise natural logarithm |
| $N(\boldsymbol{x})$ | The function counting the number of masked tokens in the sequence $\boldsymbol{x}$ |
| $|\mathcal{X}|$ | The size of the vocabulary $\mathcal{X}$ |

**Distributions**

| | |
|---|---|
| $q$ | The continuous-time forward process |
| $\tilde{q}$ | The discrete forward process |
| $p_\theta$ | The parameterized reverse process |
| $\mathcal{U}(a, b)$ | The uniform distribution on the interval $[a, b]$ |
| $\mathcal{B}(a, b)$ | The Beta distribution with parameters $a, b > 0$ |
| $\mathcal{G}(0, 1)$ | The standard Gumbel distribution |
| $\mathcal{TG}(0, 1, M)$ | The right-truncated standard Gumbel distribution with threshold $M$ |
| $\text{Cat}(\boldsymbol{\pi})$ | The categorical distribution over the class probabilities $\boldsymbol{\pi}$ |

**Abbreviations**

| | |
|---|---|
| MDMs | Masked Diffusion Models |
| ARMs | Auto-Regressive Models |
| (N)ELBO | (Negative) Evidence Lower Bound |
| NFE | The Number of Function Evaluations |
| PPL | Perplexity |
| Gen PPL | Generative Perplexity |

## B  RELATED WORK

**Discrete Diffusion Models**   Diffusion models are originally built on discrete-time continuous-space Markov chains with Gaussian transition kernels (Sohl-Dickstein et al., 2015; Ho et al., 2020). They are later extended to continuous time with the theory of stochastic processes and score matching (Song et al., 2021c).

Discrete diffusion models arise from similar contexts of Markov chains but with discrete data space (Sohl-Dickstein et al., 2015; Hoogeboom et al., 2021b). D3PM (Austin et al., 2021) considers discrete-time Markov chains with several types of transition matrices (uniform, absorbing, discretized Gaussian) and derives the discrete-time variational objective (or ELBO), which is further extended to continuous-time Markov chain (CTMC) and the corresponding ELBO(Campbell et al., 2022). They employ the mean-parameterization to learn the reverse density $q_{0|t}$.

Another line of work (Meng et al., 2022; Lou et al., 2023) argues that D3PM implicitly learns the ratio of the marginal distributions $\frac{q_t(\hat{x})}{q_t(x)}$, which is referred to as the concrete score—a discrete analog to the score function in continuous diffusion. This ratio is proposed to be directly learned via a regression objective known as concrete score matching (Meng et al., 2022), similar to score matching in continuous diffusion. However, this approach faces challenges in practice due to the incompatibility of the $L_2$ loss and the fact that the ratio $\frac{q_t(\hat{x})}{q_t(x)}$ must be positive. To address this issue, SEDD (Lou et al., 2023) introduces the score entropy objective as a theoretically more robust surrogate, which also connects the concrete score with the continuous-time ELBO.

Though SEDD considers two types of transitions (uniform, absorb), the absorbing case (masked diffusion) is much more performant in practice. It involves adding a [MASK] token as the absorbing state and modeling the simple transitions between the mask state and unmasked states, akin to the mechanism of masked models. Recent studies (Shi et al., 2024; Sahoo et al., 2024) have further aligned the masked diffusion framework with continuous diffusion, resulting in simple and principled training and sampling recipes. This not only provides a unified understanding of masked diffusion models but also enables both theoretical and empirical advancements through improved parameterization and engineering techniques. We mainly follow their framework in this work.

**Masked Models and Order-Agnostic Auto-regressive Models**   Learning to reconstruct masked tokens (or patches) is an efficient self-supervised manner for both representation learning and generative modeling. The masked modeling paradigm, originally introduced by BERT (Devlin et al.,

2019), was not initially designed for generative purposes. BERT masks a fixed portion (15%) of tokens at random[10], which supports representation learning and language understanding rather than generating text from scratch. Similarly, the masked autoencoder (MAE) (He et al., 2022) adopts this approach for image representation learning but employs a higher masked ratio (75%).

Masked models can be generative when trained on sequences with a range of masked ratios. Mask-Predict (Ghazvininejad et al., 2019) extends the number of masked tokens seen during training in BERT and uses the following objective to train a language generation model:

$$\mathcal{L}_{\text{mask}} = -\mathbb{E}_{n \sim p(n)} \mathbb{E}_{\tilde{q}_{n|0}(\boldsymbol{x}_n | \boldsymbol{x}_0)} \left[ \sum\nolimits_{l : x_n^{(l)} = m} \boldsymbol{e}_{x_0^{(l)}}^\top \log \boldsymbol{\mu}_\theta^{(l)}(\boldsymbol{x}_n)) \right] \tag{13}$$

where $p(n)$ is the uniform distribution over the sequence length $L$. MaskGIT (Chang et al., 2022) uses a similar objective for image generation, but selects the number of masked tokens $n$ according to a mask scheduling function $\gamma(t)$: sample $t \sim \mathcal{U}(0, 1)$, and set $n = \lceil \gamma(t)L \rceil$. Both Mask-Predict and MaskGIT generate samples by parallel decoding. Compared to MDMs, these methods have less theoretical grounding in training and sampling. Specifically, there is no discussion of the ELBO (Eqn. (7)) where the likelihood weighting $\frac{1}{n}$ is necessary. Nevertheless, as discussed in Section 3.2, their objectives can still lead to the same optimal solution.

The ELBO of masked models is instead revealed in the context of order-agnostic auto-regressive models (Uria et al., 2014; Hoogeboom et al., 2021a). They factorize the model distribution as $p_\theta(\boldsymbol{x}_0) = \mathbb{E}_{\sigma \sim \mathcal{U}(S_L)} \prod_{n=1}^L p_\theta(x_0^{\sigma(n)} | \boldsymbol{x}_0^{\sigma(<n)})$ in the style of ARMs, but with an additional expectation over the index permutation $\sigma$ sampled from the uniform distribution on the set of $L$-permutations $S_L$. By applying Jensen's inequality, the ELBO can be derived as:

$$\log p_\theta(\boldsymbol{x}_0) \geq \mathbb{E}_{\sigma \sim \mathcal{U}(S_L)} \sum_{n=1}^L \log p_\theta(x_0^{\sigma(n)} | \boldsymbol{x}_0^{\sigma(<n)})$$

$$= \mathbb{E}_{\sigma \sim \mathcal{U}(S_L)} \sum_{n=1}^L \frac{1}{L-n+1} \sum_{k \in \sigma(\geq n)} \log p_\theta(x_0^{(k)} | \boldsymbol{x}_0^{\sigma(<n)}) \tag{14}$$

Here $\log p_\theta(x_0^{(k)} | \boldsymbol{x}_0^{\sigma(<n)})$ (predicted data probability given known tokens) is an equivalent expression for the cross-entropy term $\boldsymbol{e}_{x_0^{(k)}}^\top \log \boldsymbol{\mu}_\theta^{(k)}(\boldsymbol{x}_0^{\sigma(<n)})$: the cross-entropy extracts the $x_0^{(k)}$-th element, $\mu_\theta^{(k)}(\boldsymbol{x}_0^{\sigma(<n)})_{x_0^{(k)}}$, from the network prediction $\boldsymbol{\mu}_\theta^{(k)}(\boldsymbol{x}_0^{\sigma(<n)})$ as $p_\theta(x_0^{(k)} | \boldsymbol{x}_0^{\sigma(<n)})$. This can be interpreted as a masked prediction where $\boldsymbol{x}_0^{\sigma(<n)}$ (the first $n-1$ tokens) is known and $\boldsymbol{x}_0^{\sigma(\geq n)}$ (the remaining $L-n+1$ tokens) is masked and to be predicted. The cross-entropy loss is averaged over the masked positions. As the last $L-n+1$ positions in a random permutation are equivalent to $L-n+1$ random positions without permutation, this ELBO is equivalent to the ELBO in Eqn. (7).

**Training and Sampling Improvements of Diffusion Models**  Since the inception of diffusion models (Ho et al., 2020; Song et al., 2021c), numerous efforts have been undertaken to enhance their performance, leading to well-established training and sampling recipes.

Prevalent training improvements include designing noise schedules, modifying the parameterization and applying variance reduction techniques (Nichol & Dhariwal, 2021; Kingma et al., 2021). Notably, flow matching (Lipman et al., 2022) provides a theoretically equivalent variant of diffusion models by employing the straight-line diffusion paths and velocity parameterization. These techniques have been validated in likelihood training of diffusion models, achieving improved density estimation results on image benchmarks (Zheng et al., 2023b). The state-of-the-art image diffusion model, EDM (Karras et al., 2022), designs the parameterization according to their proposed preconditioning and first principles, which is deeply connected to velocity parameterization (Zheng et al., 2023b). When targeted at maximum likelihood training with the ELBO, instead of improving generation quality (such as FID of generated images), the design space is relatively limited. This is also the case in discrete diffusion for text generation as the perplexity metric is based on likelihood.

Training-free accelerations of diffusion sampling mainly focus on two aspects: reducing stochasticity in the sampling process and leveraging higher-order information. DDIM (Song et al., 2021a),

---

[10]More specifically, among the 15% tokens, 80% are replaced with the [MASK] token, 10% are replaced with random tokens, and 10% remain unchanged.

along with the extension to diffusion bridges (Zheng et al., 2024), generalizes the diffusion process to non-Markovian ones with lower levels of stochasticity, enabling faster sampling. Later works connect it to the probability flow ordinary differential equation (PF-ODE) formulations of diffusion models, and build dedicated high-order numerical differential equation solvers (Zhang & Chen, 2022; Lu et al., 2022b; Zheng et al., 2023a; Gonzalez et al., 2024).

However, adapting these sampling recipes to discrete diffusion is not feasible, as the underlying evolution process of discrete data cannot be described by an ODE. Designing effective samplers for discrete diffusion requires a specialized inspection of the reverse-time Markov chain. Previous works (Chen & Ying, 2024; Chen et al., 2023) leverage the uniformization to convert continuous-time Markov chains into discrete ones, while still requiring time discretizations or approximations of the transition time distribution. Our study is the first to demonstrate that the transition time in MDMs can be sampled analytically without hyperparameter tuning or approximation errors. Infrastructure improvements, such as quantized or sparse attention (Zhang et al., 2025a;b; 2024), can also be used to accelerate the inference of discrete diffusion models, which are beyond the scope of this work.

## C  PROOF

### C.1  PROOF OF PROPOSITION 3.1

*Proof.* Denote $n_t = N(\boldsymbol{x}_t)$ as the number of masked tokens at time $t$. According to the forward process in Eqn. (1), each token is independently masked with a probability $1 - \alpha_t$, and $n_t$ follows the Binomial distribution $B(L, 1 - \alpha_t)$. The probability mass function is

$$p_t(n_t) = \binom{L}{n_t}(1 - \alpha_t)^{n_t}\alpha_t^{L-n_t}, \quad n_t = 0, 1, \dots, L \tag{15}$$

We can rearrange the sequence NELBO in Eqn. (6) as a partition by the number of masked tokens:

$$
\begin{aligned}
\mathcal{L}_\infty^{(L)} &= \int_0^1 \frac{\alpha_t'}{1 - \alpha_t}\mathbb{E}_{q_{t|0}(\boldsymbol{x}_t|\boldsymbol{x}_0)}\left[\sum\nolimits_{l:x_t^{(l)}=m}\boldsymbol{e}_{x_0^{(l)}}^\top\log\boldsymbol{\mu}_\theta^{(l)}(\boldsymbol{x}_t, t)\right]\mathrm{d}t \\
&= \int_0^1 \frac{\alpha_t'}{1 - \alpha_t}\mathbb{E}_{p_t(n_t)}\mathbb{E}_{\tilde{q}_{n_t|0}(\boldsymbol{x}_t|\boldsymbol{x}_0)}\left[\sum\nolimits_{l:x_t^{(l)}=m}\boldsymbol{e}_{x_0^{(l)}}^\top\log\boldsymbol{\mu}_\theta^{(l)}(\boldsymbol{x}_t, t)\right]\mathrm{d}t \\
&= \sum_{n=1}^L \int_0^1 \frac{\alpha_t'}{1 - \alpha_t}p_t(n)\mathbb{E}_{\tilde{q}_{n|0}(\boldsymbol{x}_n|\boldsymbol{x}_0)}\left[\sum\nolimits_{l:x_n^{(l)}=m}\boldsymbol{e}_{x_0^{(l)}}^\top\log\boldsymbol{\mu}_\theta^{(l)}(\boldsymbol{x}_n, t)\right]\mathrm{d}t \\
&= \sum_{n=1}^L \mathbb{E}_{\tilde{q}_{n|0}(\boldsymbol{x}_n|\boldsymbol{x}_0)}\left[\sum\nolimits_{l:x_n^{(l)}=m}\boldsymbol{e}_{x_0^{(l)}}^\top\left[\underbrace{\int_0^1 \frac{\alpha_t'}{1 - \alpha_t}p_t(n)\log\boldsymbol{\mu}_\theta(\boldsymbol{x}_n, t)\mathrm{d}t}_{\text{time-related term}}\right]^{(l)}\right]
\end{aligned}
\tag{16}
$$

The time-related term can be further simplified as

$$
\begin{aligned}
&\int_0^1 \frac{\alpha_t'}{1 - \alpha_t}p_t(n)\log\boldsymbol{\mu}_\theta(\boldsymbol{x}_n, t)\mathrm{d}t \\
&= \int_0^1 \frac{\alpha_t'}{1 - \alpha_t}\binom{L}{n}(1 - \alpha_t)^n\alpha_t^{L-n}\log\boldsymbol{\mu}_\theta(\boldsymbol{x}_n, t)\mathrm{d}t \\
&= \binom{L}{n}\int_{\alpha_0}^{\alpha_1}(1 - \alpha_t)^{n-1}\alpha_t^{L-n}\log\boldsymbol{\mu}_\theta(\boldsymbol{x}_n, t)\mathrm{d}\alpha_t \\
&= -\binom{L}{n}\int_0^1(1 - \alpha_t)^{n-1}\alpha_t^{L-n}\log\boldsymbol{\mu}_\theta(\boldsymbol{x}_n, t)\mathrm{d}\alpha_t \\
&= -\binom{L}{n}\frac{(n-1)!(L-n)!}{L!}\mathbb{E}_{\alpha_n\sim\mathcal{B}(L-n+1,n)}\left[\log\boldsymbol{\mu}_\theta(\boldsymbol{x}_n, \alpha^{-1}(\alpha_n))\right] \\
&= -\frac{1}{n}\underbrace{\mathbb{E}_{\alpha_n\sim\mathcal{B}(L-n+1,n)}\left[\log\boldsymbol{\mu}_\theta(\boldsymbol{x}_n, \alpha^{-1}(\alpha_n))\right]}_{:=\log\bar{\boldsymbol{\mu}}_\theta(\boldsymbol{x}_n)}
\end{aligned}
\tag{17}
$$

which completes the proof. □

## C.2 PROOF OF PROPOSITION 3.2

*Proof.* We consider minimizing the sequence NELBO (Eqn. (6)) under the expectation of the data distribution $q_0(\boldsymbol{x}_0)$:

$$
\begin{aligned}
\min_{\theta} \mathbb{E}_{q_0(\boldsymbol{x}_0)} \mathcal{L}_{\infty}^{(L)} &= \int_0^1 \frac{\alpha_t'}{1-\alpha_t} \mathbb{E}_{q_0(\boldsymbol{x}_0)} \mathbb{E}_{q_{t|0}(\boldsymbol{x}_t|\boldsymbol{x}_0)} \left[ \sum_{l: x_t^{(l)}=m} \boldsymbol{e}_{x_0^{(l)}}^{\top} \log \boldsymbol{\mu}_{\theta}^{(l)}(\boldsymbol{x}_t, t) \right] \mathrm{d}t \\
&= \int_0^1 \frac{\alpha_t'}{1-\alpha_t} \mathbb{E}_{q_t(\boldsymbol{x}_t)} \mathbb{E}_{q_{0|t}(\boldsymbol{x}_0|\boldsymbol{x}_t)} \left[ \sum_{l: x_t^{(l)}=m} \boldsymbol{e}_{x_0^{(l)}}^{\top} \log \boldsymbol{\mu}_{\theta}^{(l)}(\boldsymbol{x}_t, t) \right] \mathrm{d}t \\
&= \int_0^1 \frac{\alpha_t'}{1-\alpha_t} \mathbb{E}_{q_t(\boldsymbol{x}_t)} \left[ \sum_{l: x_t^{(l)}=m} \mathbb{E}_{q_{0|t}(\boldsymbol{x}_0|\boldsymbol{x}_t)} \left[ \boldsymbol{e}_{x_0^{(l)}} \right]^{\top} \log \boldsymbol{\mu}_{\theta}^{(l)}(\boldsymbol{x}_t, t) \right] \mathrm{d}t
\end{aligned}
\tag{18}
$$

The objective is an aggregation of cross-entropy terms over different $t, \boldsymbol{x}_t, l$. The global minimum is achieved when each cross-entropy term is optimal:

$$
\min_{\theta} - \mathbb{E}_{q_{0|t}(\boldsymbol{x}_0|\boldsymbol{x}_t)} \left[ \boldsymbol{e}_{x_0^{(l)}} \right]^{\top} \log \boldsymbol{\mu}_{\theta}^{(l)}(\boldsymbol{x}_t, t)
\tag{19}
$$

Note that $\boldsymbol{\mu}_{\theta}^{(l)} = \mathrm{softmax}(\boldsymbol{f}_{\theta}^{(l)})$ is a set of valid class probabilities that sum to 1, and $\mathbb{E}_{q_{0|t}(\boldsymbol{x}_0|\boldsymbol{x}_t)} \left[ \boldsymbol{e}_{x_0^{(l)}} \right]$ also sum to 1. Denote them as $\boldsymbol{P}$ and $\hat{\boldsymbol{P}}$ respectively, we are essentially minimizing $-\boldsymbol{P} \log \hat{\boldsymbol{P}} = D_{\mathrm{KL}}(\boldsymbol{P} \| \hat{\boldsymbol{P}}) + H(\boldsymbol{P}) \geq H(\boldsymbol{P})$, where $D_{\mathrm{KL}}(\cdot \| \cdot)$ is the Kullback–Leibler (KL) divergence and $H(\cdot)$ is the entropy. According to the property of KL divergence, the equality holds if and only if $\hat{\boldsymbol{P}} = \boldsymbol{P}$. This implies that the optimal $\theta^*$ satisfies

$$
\boldsymbol{\mu}_{\theta^*}^{(l)}(\boldsymbol{x}_t, t) = \mathbb{E}_{q_{0|t}(\boldsymbol{x}_0|\boldsymbol{x}_t)} \left[ \boldsymbol{e}_{x_0^{(l)}} \right]
\tag{20}
$$

This expression is similar to continuous diffusion, where the optimal data predictor is $\boldsymbol{\mu}_{\theta^*}(\boldsymbol{x}_t, t) = \mathbb{E}_{q_{0|t}(\boldsymbol{x}_0|\boldsymbol{x}_t)}[\boldsymbol{x}_0]$. The key difference is that, the posterior $q_{0|t}(\boldsymbol{x}_0|\boldsymbol{x}_t)$ in MDMs only depends on $\boldsymbol{x}_t$ and is irrelevant to the time $t$. Denote $n_t$ as the number of masked tokens at time $t$, we have

$$
q_{0|t}(\boldsymbol{x}_0|\boldsymbol{x}_t) = \frac{q_0(\boldsymbol{x}_0) q_{t|0}(\boldsymbol{x}_t|\boldsymbol{x}_0)}{q_t(\boldsymbol{x}_t)} = \frac{q_0(\boldsymbol{x}_0) q_{t|0}(\boldsymbol{x}_t|\boldsymbol{x}_0)}{\sum_{\boldsymbol{x}_0} q_0(\boldsymbol{x}_0) q_{t|0}(\boldsymbol{x}_t|\boldsymbol{x}_0)} = \frac{q_0(\boldsymbol{x}_0) \mathbb{E}_{p_t(n_t)}[\tilde{q}_{n_t|0}(\boldsymbol{x}_t|\boldsymbol{x}_0)]}{\sum_{\boldsymbol{x}_0} q_0(\boldsymbol{x}_0) \mathbb{E}_{p_t(n_t)}[\tilde{q}_{n_t|0}(\boldsymbol{x}_t|\boldsymbol{x}_0)]}
\tag{21}
$$

where $p_t(n_t)$ is distribution of $n_t$ at time $t$, and $\tilde{q}_{n_t|0}$ is the discrete forward process that randomly masks $n_t$ tokens. As $\boldsymbol{x}_t$ is known, the number of masked tokens is fixed as $N(\boldsymbol{x}_t)$, and $\tilde{q}_{n_t|0}(\boldsymbol{x}_t|\boldsymbol{x}_0) = 0$ for $n_t \neq N(\boldsymbol{x}_t)$. Therefore,

$$
q_{0|t}(\boldsymbol{x}_0|\boldsymbol{x}_t) = \frac{q_0(\boldsymbol{x}_0) p_t(N(\boldsymbol{x}_t)) \tilde{q}_{N(\boldsymbol{x}_t)|0}(\boldsymbol{x}_t|\boldsymbol{x}_0)}{\sum_{\boldsymbol{x}_0} q_0(\boldsymbol{x}_0) p_t(N(\boldsymbol{x}_t)) \tilde{q}_{N(\boldsymbol{x}_t)|0}(\boldsymbol{x}_t|\boldsymbol{x}_0)} = \tilde{q}_{0|N(\boldsymbol{x}_t)}(\boldsymbol{x}_0|\boldsymbol{x}_t)
\tag{22}
$$

which completes the proof. $\qquad\square$

## C.3 PROOF OF PROPOSITION 4.1

We first present a lemma that enables sequential sampling of order statistics and supports the recursive process for sampling the first hitting time.

**Lemma C.1** (Uniform Distribution Conditioned on the Maximum). *Suppose $n$ random variables $u_1, u_2, \ldots, u_n$ are independent samples from the uniform distribution $\mathcal{U}(0, \theta)$ ($\theta > 0$). Given the condition that $u = \max\{u_1, \cdots, u_n\}$, the remaining variables $u_i$ ($u_i \neq u$) follow the distribution $\mathcal{U}(0, u)$.*

*Proof.* Without loss of generality, we derive the conditional distribution of $u_1$. Other remaining variables follow the same distribution due to symmetry. For $x \leq y \leq \theta$, we have

$$
P(u_1 \leq x, u \leq y) = P(u_1 \leq x, u_2 \leq y, \ldots u_n \leq y) = P(u_1 \leq x) \prod_{i=2}^{n} P(u_i \leq y) = \frac{xy^{n-1}}{\theta^n}
\tag{23}
$$

$$P(u_1 \leq x, u \leq y | u_1 = u) = P(u \leq x) = P(u_1 \leq x, \ldots, u_n \leq x) = \prod_{i=1}^{n} P(u_i \leq x) = \frac{x^n}{\theta^n} \quad (24)$$

and

$$P(u_1 = u) = \frac{1}{n}, \qquad P(u_1 \neq u) = \frac{n-1}{n} \quad (25)$$

Therefore,

$$
\begin{aligned}
P(u_1 \leq x, u \leq y | u_1 \neq u) &= \frac{P(u_1 \leq x, u \leq y) - P(u_1 = u)P(u_1 \leq x, u \leq y | u_1 = u)}{P(u_1 \neq u)} \\
&= \frac{n}{n-1} \frac{xy^{n-1}}{\theta^n} - \frac{1}{n-1} \frac{x^n}{\theta^n}
\end{aligned}
\quad (26)
$$

By taking derivatives w.r.t. $x$ and $y$, we obtain the density $p(u_1 = x, u = y | u_1 \neq u) = \frac{ny^{y-2}}{\theta^n}$. Similarly, $P(u \leq y) = \frac{y^n}{\theta^n}$, and the density $p(u = y) = \frac{ny^{n-1}}{\theta^n}$. Therefore

$$p(u_1 = x | u = y, u_1 \neq u) = \frac{p(u_1 = x, u = y | u_1 \neq u)}{p(u = y)} = \frac{1}{y} \quad (27)$$

We conclude that $u_1$ ($u_1 \neq u$) follows a uniform distribution over the interval $[0, u]$. $\qquad \square$

Then we prove Proposition 4.1 below.

*Proof.* We first consider the case of a single token undergoing the reverse process described in Eqn. (10). Starting from time $t$, when $x_t = m$ is the mask token, we denote $\tau$ as the time at which the unmasking transition occurs (i.e., $x_{\tau+\mathrm{d}t} = m$ and $x_\tau \neq m$). The transition time $\tau$ is a random variable, whose cumulative distribution function (CDF) is available:

$$P(\tau \leq s) = p_\theta(x_s = m | x_t = m) = \frac{1 - \alpha_s}{1 - \alpha_t} \quad (28)$$

Therefore, using *inverse transform sampling*, $\tau$ can be analytically sampled by (1) drawing $u \sim \mathcal{U}(0, 1)$, and (2) solving the equation $\frac{1 - \alpha_\tau}{1 - \alpha_t} = u$.

Next, we consider the case of multiple tokens in a sequence of length $L$. Thanks to the theoretical assumptions of MDMs, the transition times of different tokens are independent in the reverse process. However, to enable token-by-token decoding, we need to sample the $L$ transition times in descending order, i.e., $1 > \tau_{L-1} > \cdots > \tau_0$. Starting from $t = 1$ with $\alpha_1 = 0$, each transition time $\tau$ can be sampled by drawing $u \sim \mathcal{U}(0, 1)$ and solving $1 - \alpha_\tau = u$ according to the single token case. In order to sample $\tau$ sequentially, we are essentially drawing the order statistics $u_{(L-1)} > \cdots > u_{(0)}$ of $L$ independent uniform variables on $[0, 1]$.

According to Lemma C.1, this process can be conducted in a recursive manner without sorting. Suppose there are currently $n$ remaining masked tokens, and the most recent unmasking occurred at time $\tau_n$. The transition time $\tau_n$ corresponds to the $n$-th smallest uniform variable $u_{(n)}$ through the relation $1 - \alpha_{\tau_n} = u_{(n)}$. To obtain the next transition time $\tau_{n-1}$, we need to sample the next order statistic $u_{(n-1)}$. By recursively applying Lemma C.1, we know that the remaining $n$ smallest uniform variables follow the distribution $\mathcal{U}(0, u_{(n)})$, if not considering their relative order. Furthermore, $u_{(n-1)}$, as the maximum of these $n$ variables, has the CDF $P(u_{(n-1)} \leq x) = \frac{x^n}{u_{(n)}^n}$ and can be sampled by solving $\frac{u_{(n-1)}^n}{u_{(n)}^n} = u_n$, where $u_n \sim \mathcal{U}(0, 1)$ (using inverse transform sampling). Therefore, the next transition time $\tau_{n-1}$ satisfies

$$1 - \alpha_{\tau_{n-1}} = u_{(n-1)} = u_{(n)} u_n^{\frac{1}{n}} = (1 - \alpha_{\tau_n}) u_n^{\frac{1}{n}} \quad (29)$$

which is equivalent to Eqn. (10) using the inverse noise schedule function $\tau_{n-1} = \alpha^{-1}(\alpha_{\tau_{n-1}})$. $\quad \square$

## C.4 PROOF OF PROPOSITION 5.2

*Proof.* The truncated standard Gumbel distribution $\mathcal{TG}(0, 1, M)$ has the probability density function (PDF) and cumulative distribution function (CDF) defined as follows:

$$\hat{f}(x) = \frac{f(x)}{F(M)} \mathbb{I}_{x \leq M}, \quad \hat{F}(x) = \min\left\{ \frac{F(x)}{F(M)}, 1 \right\} \tag{30}$$

where

$$f(x) = e^{-x - e^{-x}}, \quad F(x) = e^{-e^{-x}} \tag{31}$$

are the PDF and CDF of the standard Gumbel distribution $\mathcal{G}(0, 1)$, and $M$ is the right truncation point. Suppose the class probabilities are sorted as $\pi_1 \leq \cdots \leq \pi_K$, and denote $\pi_0 = 0, \theta_n = \log \pi_n$ for simplicity. To conduct truncated Gumbel-based categorical sampling, $K$ i.i.d. samples $\{g_i\}_{i=1}^K$ are drawn from $\mathcal{TG}(0, 1, M)$. The resulting categorical probability of class $n$ is

$$
\begin{aligned}
P(\mathrm{argmax}(\theta_i + g_i) = n) &= \int_{-\infty}^{+\infty} \hat{f}(g) \prod_{k \neq n} P(\theta_k + g_k \leq \theta_n + g) \mathrm{d}g \\
&= \int_{-\infty}^{M} \hat{f}(g) \prod_{k \neq n} \hat{F}(\theta_n + g - \theta_k) \mathrm{d}g \\
&= e^{Ke^{-M}} \int_{-\infty}^{M} e^{-g - e^{-g}} \prod_{k \neq n} \min\{ e^{-e^{-M}}, e^{-e^{-\theta_n - g + \theta_k}} \} \mathrm{d}g \\
&= e^{Ke^{-M}} \int_{-\infty}^{M} e^{-g - e^{-g}} e^{-\sum_{k \neq n} e^{-\min\{g + \theta_n - \theta_k, M\}}} \mathrm{d}g \\
&= e^{Ke^{-M}} \sum_{i=1}^{n} \int_{\theta_{i-1} + M - \theta_n}^{\theta_i + M - \theta_n} e^{-g} e^{-(\sum_{k=i}^{K} e^{\theta_k - \theta_n}) e^{-g}} e^{-(i-1)e^{-M}} \mathrm{d}g
\end{aligned} \tag{32}
$$

where the integral has a closed-form solution by

$$\int e^{-g} e^{-Ae^{-g}} \mathrm{d}g = \frac{e^{-Ae^{-g}}}{A} + C \tag{33}$$

With this, Eqn. (32) can be further simplified to

$$
\begin{aligned}
&P(\mathrm{argmax}(\theta_i + g_i) = n) \\
=& e^{Ke^{-M}} \sum_{i=1}^{n} \frac{e^{-(\sum_{k=i}^{K} e^{\theta_k - \theta_n}) e^{\theta_n - \theta_i - M}} - e^{-(\sum_{k=i}^{K} e^{\theta_k - \theta_n}) e^{\theta_n - \theta_{i-1} - M}}}{\sum_{k=i}^{K} e^{\theta_k - \theta_n}} e^{-(i-1)e^{-M}} \\
=& e^{Ke^{-M}} \pi_n \sum_{i=1}^{n} \frac{e^{-\frac{\sum_{k=i}^{K} \pi_k}{\pi_i} e^{-M}} - e^{-\frac{\sum_{k=i}^{K} \pi_k}{\pi_{i-1}} e^{-M}}}{\sum_{k=i}^{K} \pi_k} e^{-(i-1)e^{-M}} \\
=& \pi_n \sum_{i=1}^{n} \frac{e^{\left( K+1-i-\frac{\sum_{k=i}^{K} \pi_k}{\pi_i} \right) e^{-M}} - e^{\left( K+1-i-\frac{\sum_{k=i}^{K} \pi_k}{\pi_{i-1}} \right) e^{-M}}}{\sum_{k=i}^{K} \pi_k}
\end{aligned} \tag{34}
$$

Therefore, the original class probabilities $\{\pi_n\}_{n=1}^K$ are shifted to $\{\pi'_n\}_{n=1}^K$ if the Gumbel variables used in categorical sampling are right-truncated to $M$. $\pi'_n$ is given by

$$
\begin{aligned}
\pi'_n &= \pi_n \sum_{i=1}^{n} \frac{e^{\left( K+1-i-\frac{\sum_{k=i}^{K} \pi_k}{\pi_i} \right) e^{-M}} - e^{\left( K+1-i-\frac{\sum_{k=i}^{K} \pi_k}{\pi_{i-1}} \right) e^{-M}}}{\sum_{k=i}^{K} \pi_k} \\
&= \pi_n \sum_{i=1}^{n} \frac{e^{\left( K-i-\frac{\sum_{k=i+1}^{K} \pi_k}{\pi_i} \right) e^{-M}} - e^{\left( K-(i-1)-\frac{\sum_{k=i}^{K} \pi_k}{\pi_{i-1}} \right) e^{-M}}}{\sum_{k=i}^{K} \pi_k}
\end{aligned} \tag{35}
$$

We can verify that $\{\pi'_n\}_{n=1}^K$ are valid class probabilities that sum to 1:

$$
\begin{aligned}
\sum_{n=1}^{K} \pi'_n &= \sum_{n=1}^{K} \pi_n \sum_{i=1}^{n} \frac{e^{\left(K-i-\frac{\sum_{k=i+1}^{K}\pi_k}{\pi_i}\right)e^{-M}} - e^{\left(K-(i-1)-\frac{\sum_{k=i}^{K}\pi_k}{\pi_{i-1}}\right)e^{-M}}}{\sum_{k=i}^{K}\pi_k} \\
&= \sum_{i=1}^{K}\sum_{n=i}^{K} \pi_n \frac{e^{\left(K-i-\frac{\sum_{k=i+1}^{K}\pi_k}{\pi_i}\right)e^{-M}} - e^{\left(K-(i-1)-\frac{\sum_{k=i}^{K}\pi_k}{\pi_{i-1}}\right)e^{-M}}}{\sum_{k=i}^{K}\pi_k} \\
&= \sum_{i=1}^{K} e^{\left(K-i-\frac{\sum_{k=i+1}^{K}\pi_k}{\pi_i}\right)e^{-M}} - e^{\left(K-(i-1)-\frac{\sum_{k=i}^{K}\pi_k}{\pi_{i-1}}\right)e^{-M}} \\
&= e^{(K-K)e^{-M}} - e^{\left(K-\frac{1}{\pi_0}\right)e^{-M}} = 1
\end{aligned}
\tag{36}
$$

where $\pi_0 = 0$ and $e^{\left(K-\frac{1}{\pi_0}\right)e^{-M}} = 0$. $\qquad\square$

## D  RELATIONSHIP BETWEEN MASKED DIFFUSION MODELS AND PREVIOUS DISCRETE DIFFUSION MODELS

Our framework and notations are based on recent studies of MDMs (Shi et al., 2024; Sahoo et al., 2024), which offer a theoretically simplified and empirically improved version of the best-performing absorbing case in discrete diffusion models (Austin et al., 2021; Campbell et al., 2022; Lou et al., 2023). In this section, we present a summary of some background information on previous formulations: generative modeling of discrete data via continuous-time Markov chains (Section D.1), robust and principled training and sampling with score parameterization (Section D.2), and their equivalence to MDMs (Section D.3).

### D.1  DISCRETE DIFFUSION VIA CONTINUOUS-TIME MARKOV CHAINS

Continuous-time Markov chains (CTMCs) (Anderson, 2012) are a fundamental concept in stochastic processes used to model systems that transition between discrete states continuously over time.

**Forward Process**  Denote $\mathcal{X}$ as the state space and $x \in \mathcal{X}$ as a state. The probability of transitioning from one state $x$ to another state $\hat{x}$ near time $t$ is governed by the transition rate matrix $\boldsymbol{Q}_t \in \mathbb{R}^{|\mathcal{X}|\times|\mathcal{X}|}$. Specifically, denote $Q_t(x,\hat{x})$ as the transition rate from $x$ to $\hat{x}$, the transition probability during a small time interval $\Delta t$ is

$$
p_{t+\Delta t|t}(\hat{x}|x) = \delta_{x,\hat{x}} + Q_t(x,\hat{x})\Delta t + \mathcal{O}((\Delta t)^2)
\tag{37}
$$

The off-diagonal elements $Q_t(x,\hat{x})$ $(x \neq \hat{x})$ are non-negative, and the diagonal elements $Q_t(x,x) = -\sum_{\hat{x}\neq x} Q_t(x,\hat{x}) \leq 0$, ensuring that each row of $\boldsymbol{Q}_t$ sums to zero (so that $p_t$ does not gain or lose total mass). Equivalently, the transition rate can be defined by the transition probability as

$$
Q_t(x,\hat{x}) = \lim_{\Delta t \to 0} \frac{p_{t+\Delta t|t}(\hat{x}|x) - \delta_{x,\hat{x}}}{\Delta t}
\tag{38}
$$

Denote $\boldsymbol{p}_t = \{p_t(x)\}_{x\in\mathcal{X}}$ as the marginal distributions of all states at time $t$, and $\boldsymbol{P}_{t|s} \in \mathbb{R}^{|\mathcal{X}|\times|\mathcal{X}|}$ as the forward transition matrix from time $s$ to time $t$ satisfying $P_{t|s}(x,\hat{x}) = p_{t|s}(\hat{x}|x)$. The Kolmogorov forward (or Fokker-Planck) equations describe the evolution of both the marginals $\boldsymbol{p}_t$ (starting from the data distribution) and the transition matrix $\boldsymbol{P}_{t|s}$:

$$
\frac{d\boldsymbol{p}_t}{dt} = \boldsymbol{p}_t\boldsymbol{Q}_t, \quad \frac{d\boldsymbol{P}_{t|s}}{dt} = \boldsymbol{P}_{t|s}\boldsymbol{Q}_t
\tag{39}
$$

In practice, the forward process is designed to be simple degradation (Campbell et al., 2022; Lou et al., 2023), such that $p_t$ approaches a stationary distribution $p_{\text{base}}$ that is easy to sample from as $t$ increases, akin to the Gaussian noising process in continuous diffusion. Specifically, the transition rate matrix $\boldsymbol{Q}_t$ is set to $\sigma(t)\boldsymbol{Q}$ where $\sigma(t)$ is a scalar noise schedule function and $\boldsymbol{Q}$ is a constant

matrix with low ranks. In this case, the transition matrix can be solved analytically as $\boldsymbol{P}_{t|s} = e^{(\bar{\sigma}(t)-\bar{\sigma}(s))\boldsymbol{Q}}$ where $\bar{\sigma}(t) = \int_0^t \sigma(\tau)\mathrm{d}\tau$. Common choices of $\boldsymbol{Q}$ (Lou et al., 2023) include:

$$
\boldsymbol{Q}_{\text{uniform}} = \begin{bmatrix} 1-N & 1 & \cdots & 1 \\ 1 & 1-N & \cdots & 1 \\ \vdots & \vdots & \ddots & \vdots \\ 1 & 1 & \cdots & 1-N \end{bmatrix}, \quad \boldsymbol{Q}_{\text{absorb}} = \begin{bmatrix} -1 & 0 & \cdots & 0 & 1 \\ 0 & -1 & \cdots & 0 & 1 \\ \vdots & \vdots & \ddots & \vdots & \vdots \\ 0 & 0 & \cdots & -1 & 1 \\ 0 & 0 & \cdots & 0 & 0 \end{bmatrix} \tag{40}
$$

The former disturbs the data distribution into a uniform one, and the latter additionally adds a [MASK] token as the absorbing state.

**Time Reversal** Similar to continuous diffusion, discrete diffusion defined above has a time reversal (Kelly, 2011; Sun et al., 2022) described by the reverse transition rate matrix $\bar{\boldsymbol{Q}}_t$ which satisfies

$$
\bar{Q}_t(x, \hat{x}) = \begin{cases} \dfrac{p_t(\hat{x})}{p_t(x)} Q_t(\hat{x}, x), & \hat{x} \neq x \\ -\sum_{y \neq x} \bar{Q}_t(x, y), & \hat{x} = x \end{cases} \tag{41}
$$

The intractable ratio $\frac{p_t(\hat{x})}{p_t(x)}$, named concrete score (Meng et al., 2022), acts as an analog to the score function (Song et al., 2021c) in continuous diffusion. The reverse process can be described as

$$
\frac{\mathrm{d}\boldsymbol{p}_s}{\mathrm{d}s} = -\boldsymbol{p}_s \bar{\boldsymbol{Q}}_s, \quad \frac{\mathrm{d}\boldsymbol{P}_{s|t}}{\mathrm{d}s} = -\boldsymbol{P}_{s|t} \bar{\boldsymbol{Q}}_s \tag{42}
$$

which evolves backward in time with $s$ decreasing to 0 and $s < t$. It is sufficient to simulate the whole process as long as the concrete score, the only unknown term in $\bar{\boldsymbol{Q}}_t$, is estimated.

### D.2 Score-Entropy Discrete Diffusion (SEDD)

SEDD (Lou et al., 2023) provides principled, robust and scalable techniques for score-based training and sampling of discrete diffusion models.

**Parameterization** SEDD parameterizes a score prediction network $\boldsymbol{s}_\theta(x, t) \in \mathbb{R}^{|\mathcal{X}|}$ to learn the unknown concrete score $\left\{ \frac{p_t(\hat{x})}{p_t(x)} \right\}_{\hat{x} \in \mathcal{X}}$. We use $s_\theta(x, t)_y$ to denote its $y$-th element.

**Training Objective** SEDD proposes the diffusion-weighted denoising score entropy (DWDSE) objective to optimize $\boldsymbol{s}_\theta(x, t)$:

$$
\mathcal{L}_{\text{DWDSE}}(x_0) = \int_0^T \mathbb{E}_{x_t \sim p_{t|0}(\cdot|x_0)} \sum_{\hat{x}_t \neq x_t} Q_t(\hat{x}_t, x_t) I\left( s_\theta(x_t, t)_{\hat{x}_t}, \frac{p_{t|0}(\hat{x}_t \mid x_0)}{p_{t|0}(x_t \mid x_0)} \right) \mathrm{d}t \tag{43}
$$

where $I(a, b) := a - b \log a + K(b)$, and $K(b) := b \log b - b$ is a normalizing constant function that ensures $I(a, b) \geq 0$. Eqn. (43) not only admits the optimal solution as the concrete score, but also serves as a NELBO for discrete diffusion models by $-\log p_0^\theta(\boldsymbol{x}_0) \leq \mathcal{L}_{\text{DWDSE}}(x_0) + D_{\text{KL}}(p_{T|0}(\cdot|x_0) \| p_{\text{base}})$, where $p_{\text{base}}$ is the stationary distribution when $T \to \infty$.

**Sampling Procedures** Denote $\boldsymbol{s}_t(x) = \left\{ \frac{p_t(\hat{x})}{p_t(x)} \right\}_{\hat{x} \in \mathcal{X}}$ as the ground-truth concrete score, and $s_t(x)_y$ as its $y$-th element. We use $\boldsymbol{s}_t(x)$ to demonstrate the sampling process, while in practice it is replaced with the learned score $\boldsymbol{s}_\theta(x, t)$. SEDD offers two sampling procedures: Euler sampling and analytic sampling with Tweedie $\tau$-Leaping.

Euler sampling applies the Euler discretization to the reverse process (Eqn. (42)), producing a reverse transition similar to the forward transition in Eqn. (37):

$$
p_{s|t}^{\text{Euler}}(x_s|x_t) = \delta_{x_t, x_s} + \bar{Q}_t(x_t, x_s)(t - s) \tag{44}
$$

It can be expressed by the concrete score $s_t$ as

$$p_{s|t}^{\text{Euler}}(x_s|x_t) = \begin{cases} (t-s)Q_t(x_s, x_t)s_t(x_t)_{x_s}, & x_s \neq x_t \\ 1 - \sum_{y \neq x_t} p_{s|t}^{\text{Euler}}(y|x_t), & x_s = x_t \end{cases} \tag{45}$$

The Euler sampling implicitly assumes a constant reverse rate matrix $\bar{\boldsymbol{Q}}_\tau = \bar{\boldsymbol{Q}}_t$ for $\tau \in [s, t]$, producing approximation errors and even resulting in negative probabilities at $x_s = x_t$.

Tweedie $\tau$-leaping operates similarly to the posterior sampling in DDPM (Ho et al., 2020), by analytically solving the posterior $p_{s|t}(x_s|x_t)$ given the ground-truth concrete score. Specifically,

$$p_{s|t}^{\text{Tweedie}}(x_s|x_t) = \frac{p_{t|s}(x_t|x_s)p_s(x_s)}{p_t(x_t)} \tag{46}$$

Under the special choice $\boldsymbol{Q}_t = \sigma(t)\boldsymbol{Q}$ described in the previous section, we have

$$p_{t|s}(x_t|x_s) = \left(\boldsymbol{P}_{t|s}\right)_{x_s, x_t} = \left(e^{(\bar{\sigma}(t) - \bar{\sigma}(s))\boldsymbol{Q}}\right)_{x_s, x_t} \tag{47}$$

and

$$p_s(x_s) = (\boldsymbol{p}_s)_{x_s} = \left(\boldsymbol{p}_t \boldsymbol{P}_{t|s}^{-1}\right)_{x_s} = \left(\boldsymbol{p}_t e^{-(\bar{\sigma}(t) - \bar{\sigma}(s))\boldsymbol{Q}}\right)_{x_s} \tag{48}$$

Therefore,

$$\begin{aligned} p_{s|t}^{\text{Tweedie}}(x_s|x_t) &= \left(e^{(\bar{\sigma}(t) - \bar{\sigma}(s))\boldsymbol{Q}}\right)_{x_s, x_t} \left(\frac{\boldsymbol{p}_t}{p_t(x_t)} e^{-(\bar{\sigma}(t) - \bar{\sigma}(s))\boldsymbol{Q}}\right)_{x_s} \\ &= \left(e^{(\bar{\sigma}(t) - \bar{\sigma}(s))\boldsymbol{Q}}\right)_{x_s, x_t} \left(\boldsymbol{s}_t(x_t) e^{-(\bar{\sigma}(t) - \bar{\sigma}(s))\boldsymbol{Q}}\right)_{x_s} \end{aligned} \tag{49}$$

**Multi-Dimensional Case**  For a token sequence $\boldsymbol{x} \in \mathcal{X}^L$ of length $L$, we use $-l$ to denote the indexes of all tokens except the $l$-th one. The concrete score $s_t(\boldsymbol{x}) \in \mathbb{R}^{|\mathcal{X}| \times L}$ is defined between sequences that differ by a Hamming distance of 1:

$$s_t(\boldsymbol{x})_{\hat{x}, l} = \frac{p_t(\hat{\boldsymbol{x}})}{p_t(\boldsymbol{x})}, \quad \text{s.t. } \hat{x}^{(l)} = \hat{x}, \hat{\boldsymbol{x}}^{(-l)} = \boldsymbol{x}^{(-l)} \tag{50}$$

The forward and reverse processes are factorized across dimensions in the same manner as in MDMs described in the main text. The parameterized score network $\boldsymbol{s}_\theta(\boldsymbol{x}, t) \in \mathbb{R}^{|\mathcal{X}| \times L}$ also predicts the scores at all positions at a time. Consequently, both the training and sampling are conducted simultaneously and independently for all dimensions, except that the network input $\boldsymbol{x}$ contains the current sequence information.

### D.3 Connection between MDMs and SEDD Absorb

The absorbing case ($\boldsymbol{Q} = \boldsymbol{Q}_{\text{absorb}}$) of discrete diffusion has demonstrated both simple formulations and superior performance. As revealed in previous and concurrent works (Shi et al., 2024; Ou et al., 2024), SEDD Absorb is theoretically equivalent to MDMs in multiple aspects. For simplicity, we focus on the single-token case, as the factorization approach used in both MDMs and SEDD Absorb ensures that equivalence in one dimension implies equivalence in multiple dimensions.

#### D.3.1 Equivalence of Training

**Relation between Forward Processes**  The forward transition matrix of SEDD Absorb is $\boldsymbol{P}_{t|0} = e^{\bar{\sigma}(t)\boldsymbol{Q}_{\text{absorb}}}$. It can be verified by mathematical induction that $\boldsymbol{Q}_{\text{absorb}}^n = (-1)^{n-1}\boldsymbol{Q}_{\text{absorb}}$ for any positive integer $n$. With this identity, $\boldsymbol{P}_{t|0}$ can be simplified as

$$\boldsymbol{P}_{t|0} = e^{\bar{\sigma}(t)\boldsymbol{Q}_{\text{absorb}}} = \boldsymbol{I} + \sum_{k=1}^{\infty} \frac{\bar{\sigma}(t)^k \boldsymbol{Q}_{\text{absorb}}^k}{k!} = \boldsymbol{I} - \sum_{k=1}^{\infty} \frac{(-\bar{\sigma}(t))^k}{k!} \boldsymbol{Q}_{\text{absorb}}$$

$$= \boldsymbol{I} + \left(1 - e^{-\bar{\sigma}(t)}\right) \boldsymbol{Q}_{\text{absorb}} \tag{51}$$

This is equivalent to the forward process (Eqn. (1)) in MDMs with the relation $\alpha_t = e^{-\bar{\sigma}(t)}$.

**Relation between Parameterizations** For coherence, we use $m$ to denote the [MASK] token. We only need to consider the concrete score $\boldsymbol{s}_t(m)$ at $m$, since $\boldsymbol{s}_t(x)$ for $x \neq m$ can be converted from $\boldsymbol{s}_t(m)$ by $s_t(x)_{\hat{x}} = \frac{s_t(m)_{\hat{x}}}{s_t(m)_x}$. We have

$$s_t(m)_x = \frac{p_t(x)}{p_t(m)} = \frac{\left(\boldsymbol{p}_0 \boldsymbol{P}_{t|0}\right)_x}{\left(\boldsymbol{p}_0 \boldsymbol{P}_{t|0}\right)_m} \tag{52}$$

Substituting the expression of $\boldsymbol{P}_{t|0}$ in Eqn. (51) into Eqn. (52), for $x \neq m$, we have

$$\left(\boldsymbol{p}_0 \boldsymbol{P}_{t|0}\right)_m = \sum_{x_0 \in \mathcal{X} \backslash \{m\}} p_0(x_0) p_{t|0}(m|x_0) = (1 - \alpha_t) \sum_{x_0 \in \mathcal{X} \backslash \{m\}} p_0(x_0) = 1 - \alpha_t \tag{53}$$

$$\left(\boldsymbol{p}_0 \boldsymbol{P}_{t|0}\right)_x = \sum_{x_0 \in \mathcal{X} \backslash \{m\}} p_0(x_0) p_{t|0}(x|x_0) = \sum_{x_0 \in \mathcal{X} \backslash \{m\}} p_0(x_0) \alpha_t \delta_{x_0, x} = \alpha_t p_0(x) \tag{54}$$

and

$$s_t(m)_x = \frac{\left(\boldsymbol{p}_0 \boldsymbol{P}_{t|0}\right)_x}{\left(\boldsymbol{p}_0 \boldsymbol{P}_{t|0}\right)_m} = \frac{\alpha_t}{1 - \alpha_t} p_0(x) = \frac{\alpha_t}{1 - \alpha_t} \left(\mathbb{E}_{p_{0|t}(x_0|m)}\left[\boldsymbol{e}_{x_0}\right]\right)_x \tag{55}$$

This implies that the score parameterization is related to the mean parameterization $\boldsymbol{\mu}_\theta(x, t)$ in MDMs (excluding the $m$-th dimension) by

$$\boldsymbol{s}_\theta(x, t) = \frac{\alpha_t}{1 - \alpha_t} \boldsymbol{\mu}_\theta(x, t) \tag{56}$$

**Equivalence of ELBOs** Substituting this relation between $\boldsymbol{s}_\theta$ and $\boldsymbol{\mu}_\theta$ into the score entropy objective of SEDD (Eqn. 43), we have

$$\mathcal{L}_{\text{DWDSE}}(x_0) = \int_0^T \mathbb{E}_{x_t \sim p_{t|0}(\cdot|x_0)} \left[ \delta_{x_t, m} \sum_{x \neq m} Q_t(x, m) I\left(s_\theta(m, t)_x, \frac{p_{t|0}(x \mid x_0)}{p_{t|0}(m \mid x_0)}\right) \right] dt$$

$$= \int_0^T \sigma(t) \mathbb{E}_{x_t \sim p_{t|0}(\cdot|x_0)} \left[ \delta_{x_t, m} \sum_{x \neq m} I\left(\frac{\alpha_t}{1 - \alpha_t} \mu_\theta(m, t)_x, \delta_{x, x_0} \frac{\alpha_t}{1 - \alpha_t}\right) \right] dt \tag{57}$$

Observing that $I(a, 0) = a$, $\sum_{x \neq m} \mu_\theta(m, t)_x = 1$, we have

$$\sum_{x \neq m} I\left(\frac{\alpha_t}{1 - \alpha_t} \mu_\theta(m, t)_x, \delta_{x, x_0} \frac{\alpha_t}{1 - \alpha_t}\right)$$

$$= K\left(\frac{\alpha_t}{1 - \alpha_t}\right) - \frac{\alpha_t}{1 - \alpha_t} \log\left(\frac{\alpha_t}{1 - \alpha_t} \mu_\theta(m, t)_{x_0}\right) + \frac{\alpha_t}{1 - \alpha_t} \sum_{x \neq m} \mu_\theta(m, t)_x \tag{58}$$

$$= -\frac{\alpha_t}{1 - \alpha_t} \log \mu_\theta(m, t)_{x_0}$$

Using $\alpha_t' = -\sigma(t)\alpha_t$, the objective $\mathcal{L}_{\text{DWDSE}}(x_0)$ can be simplified to

$$\mathcal{L}_{\text{DWDSE}}(x_0) = -\int_0^T \sigma(t) \mathbb{E}_{x_t \sim p_{t|0}(\cdot|x_0)} \left[ \delta_{x_t, m} \frac{\alpha_t}{1 - \alpha_t} \log \mu_\theta(m, t)_{x_0} \right] dt$$

$$= \int_0^T \frac{\alpha_t'}{1 - \alpha_t} \mathbb{E}_{x_t \sim p_{t|0}(\cdot|x_0)} \left[ \delta_{x_t, m} \log \mu_\theta(x_t, t)_{x_0} \right] dt \tag{59}$$

As $\log \mu_\theta(x_t, t)_{x_0}$ and $\boldsymbol{e}_{x_0}^\top \log \boldsymbol{\mu}_\theta(x_t, t)$ are equivalent expressions for the cross entropy, we conclude that $\mathcal{L}_{\text{DWDSE}}(x_0)$ is equal to the NELBO for MDMs (Eqn. (5)) when $T = 1$.

### D.3.2 EQUIVALENCE OF SAMPLING

**Euler Sampler and Tweedie $\tau$-Leaping Sampler in SEDD** are equivalent in the absorbing case under the linear noise schedule $\alpha_t = e^{-\bar{\sigma}(t)} = 1 - t$.

On the one hand, the Euler sampler (Eqn. (45)) with score parameterization network $s_\theta$ is

$$p_{s|t}^{\text{Euler}}(x_s|x_t) = \begin{cases} (t-s)\sigma(t)\left(\boldsymbol{Q}_{\text{absorb}}\right)_{x_s,x_t} s_\theta(x_t,t)_{x_s}, & x_s \neq x_t \\ 1 - \sum_{y \neq x_t} p_{s|t}^{\text{Euler}}(y|x_t), & x_s = x_t \end{cases} \tag{60}$$

When $x_s \neq x_t$, using the identities $s_\theta(x,t)_x = 1$ and $\left(\boldsymbol{Q}_{\text{absorb}}\right)_{x_s,x_t} = \delta_{m,x_t} - \delta_{x_s,x_t}$, $p_{s|t}^{\text{Euler}}$ can be simplified to

$$p_{s|t}^{\text{Euler}}(x_s|x_t) = \begin{cases} 0, & x_t \neq m \\ (t-s)\,\sigma(t)s_\theta(x_t,t)_{x_s}, & x_t = m \end{cases} \tag{61}$$

When $x_s = x_t$, from Eqn. (56) we know $\sum_{x \in \mathcal{X} \setminus \{m\}} s_\theta(m,t)_x = \frac{\alpha_t}{1-\alpha_t} \sum_{x \in \mathcal{X} \setminus \{m\}} \mu_\theta(m,t)_x = \frac{\alpha_t}{1-\alpha_t}$. Hence, $p_{s|t}^{\text{Euler}}$ can be calculated as

$$p_{s|t}^{\text{Euler}}(x_s|x_t) = \begin{cases} 1, & x_t \neq m \\ 1 - \frac{\sigma(t)e^{-\bar{\sigma}(t)}(t-s)}{1-e^{-\bar{\sigma}(t)}}, & x_t = m \end{cases} \tag{62}$$

Combining Eqn. (61) and Eqn. (62), the Euler sampler is simplified to

$$p_{s|t}^{\text{Euler}}(x_s|x_t) = \begin{cases} \delta_{x_s,x_t}, & x_t \neq m \\ (t-s)\,\sigma(t)s_\theta(x_t,t)_{x_s}, & x_t = m, x_s \neq m \\ 1 - \frac{\sigma(t)e^{-\bar{\sigma}(t)}(t-s)}{1-e^{-\bar{\sigma}(t)}}, & x_t = m, x_s = m \end{cases} \tag{63}$$

On the other hand, the Tweedie $\tau$-Leaping sampler (Eqn. (49)) is

$$p_{s|t}^{\text{Tweedie}}(x_s|x_t) = \left(e^{(\bar{\sigma}(t)-\bar{\sigma}(s))\boldsymbol{Q}}\right)_{x_s,x_t} \left(\boldsymbol{s}_\theta(x_t,t)e^{-(\bar{\sigma}(t)-\bar{\sigma}(s))\boldsymbol{Q}}\right)_{x_s} \tag{64}$$

When $\boldsymbol{Q} = \boldsymbol{Q}_{\text{absorb}}$, similar to Eqn. (51), we have

$$\begin{aligned} e^{(\bar{\sigma}(t)-\bar{\sigma}(s))\boldsymbol{Q}} &= \boldsymbol{I} + \left(1 - e^{-(\bar{\sigma}(t)-\bar{\sigma}(s))}\right)\boldsymbol{Q}_{\text{absorb}} \\ e^{-(\bar{\sigma}(t)-\bar{\sigma}(s))\boldsymbol{Q}} &= \boldsymbol{I} + \left(1 - e^{\bar{\sigma}(t)-\bar{\sigma}(s)}\right)\boldsymbol{Q}_{\text{absorb}} \end{aligned} \tag{65}$$

Using the identities $s_\theta(x,t)_x = 1$ and $\left(\boldsymbol{Q}_{\text{absorb}}\right)_{x_s,x_t} = \delta_{m,x_t} - \delta_{x_s,x_t}$, we have

$$\begin{aligned} \left(e^{(\bar{\sigma}(t)-\bar{\sigma}(s))\boldsymbol{Q}}\right)_{x_s,x_t} &= \delta_{x_s,x_t} + \left(1 - e^{-(\bar{\sigma}(t)-\bar{\sigma}(s))}\right)\left(\boldsymbol{Q}_{\text{absorb}}\right)_{x_s,x_t} \\ &= e^{-(\bar{\sigma}(t)-\bar{\sigma}(s))}\delta_{x_s,x_t} + \left(1 - e^{-(\bar{\sigma}(t)-\bar{\sigma}(s))}\right)\delta_{m,x_t} \end{aligned} \tag{66}$$

and

$$\sum_{y \in \mathcal{X}} s_\theta(x,t)_y = \sum_{y \in \mathcal{X}} \frac{s_\theta(m,t)_y}{s_\theta(m,t)_x} = s_\theta(x,t)_m \left(s_\theta(m,t)_m + \sum_{x \in \mathcal{X} \setminus \{m\}} s_\theta(m,t)_x\right) = \frac{s_\theta(x,t)_m}{1-\alpha_t} \tag{67}$$

Therefore,

$$\begin{aligned} \left(\boldsymbol{s}_\theta(x_t,t)e^{-(\bar{\sigma}(t)-\bar{\sigma}(s))\boldsymbol{Q}}\right)_{x_s} &= s_\theta(x_t,t)_{x_s} + \left(1 - e^{\bar{\sigma}(t)-\bar{\sigma}(s)}\right)\left(\boldsymbol{s}_\theta(x_t,t)\boldsymbol{Q}_{\text{absorb}}\right)_{x_s} \\ &= s_\theta(x_t,t)_{x_s} + \left(1 - e^{\bar{\sigma}(t)-\bar{\sigma}(s)}\right)\sum_{x \in \mathcal{X}} s_\theta(x_t,t)_x \left(\boldsymbol{Q}_{\text{absorb}}\right)_{x,x_s} \\ &= e^{\bar{\sigma}(t)-\bar{\sigma}(s)}s_\theta(x_t,t)_{x_s} + \delta_{x_s,m}\left(1 - e^{\bar{\sigma}(t)-\bar{\sigma}(s)}\right)\sum_{x \in \mathcal{X}} s_\theta(x_t,t)_x \\ &= e^{\bar{\sigma}(t)-\bar{\sigma}(s)}s_\theta(x_t,t)_{x_s} + \delta_{x_s,m}\frac{1 - e^{\bar{\sigma}(t)-\bar{\sigma}(s)}}{1-e^{-\bar{\sigma}(t)}}s_\theta(x_t,t)_m \end{aligned} \tag{68}$$

Substituting Eqn. (66) and Eqn. (68) into Eqn. (64), the Tweedie $\tau$-Leaping sampler is simplified to

$$p_{s|t}^{\text{Tweedie}}(x_s|x_t) = \begin{cases} \delta_{x_s,x_t}, & x_t \neq m \\ \left(e^{\bar{\sigma}(t)-\bar{\sigma}(s)} - 1\right)s_\theta(x_t,t)_{x_s}, & x_t = m, x_s \neq m \\ \frac{1-e^{-\bar{\sigma}(s)}}{1-e^{-\bar{\sigma}(t)}}, & x_t = m, x_s = m \end{cases} \tag{69}$$

Under the linear noise schedule, as $e^{-\bar{\sigma}(t)} = 1 - t$, we have $\bar{\sigma}(t) = -\log(1-t)$ and $\sigma(t) = \frac{1}{1-t} = e^{\bar{\sigma}(t)}$. Consequently, $(t-s)\sigma(t) = e^{\bar{\sigma}(t)-\bar{\sigma}(s)} - 1$, and the Euler sampler in Eqn. (63) is the same as Tweedie $\tau$-Leaping sampler in Eqn. (69).

**Tweedie $\tau$-Leaping Sampler in SEDD and the Reverse Sampling Process in MDMs** are equivalent in the absorbing case. By the relation $\alpha_t = e^{-\bar{\sigma}(t)}$ and $s_\theta(x_t, t)_{x_s} = \frac{\alpha_t}{1-\alpha_t}\mu_\theta(x_t,t)_{x_s}(x_t = m, x_s \neq m)$, the Tweedie $\tau$-Leaping sampler in Eqn. (69) is converted to

$$p_{s|t}^{\text{Tweedie}}(x_s|x_t) = \begin{cases} \delta_{x_s,x_t}, & x_t \neq m \\ \frac{\alpha_s - \alpha_t}{1-\alpha_t}\mu_\theta(x_t,t)_{x_s}, & x_t = m, x_s \neq m \\ \frac{1-\alpha_s}{1-\alpha_t}, & x_t = m, x_s = m \end{cases} \tag{70}$$

which is the same as the reverse process in MDMs (Eqn. (3)).

# E    EXPECTED NFE IN BATCHED SAMPLING USING THE CACHING STRATEGY

Suppose the sampling is performed on timesteps $1 = t_N \to t_{N-1} \to \cdots \to t_0 = 0$, and denote $\boldsymbol{x}_t \in \mathcal{X}^{BL}$ as the concatenation of the batched sequences at time $t$. During the sampling step $t_i \to t_{i-1}$, the NFE increases by 1 when $\boldsymbol{x}_{t_{i-1}} \neq \boldsymbol{x}_{t_i}$, and remains the same otherwise. Therefore, the expected NFE (E-NFE) can be expressed by

$$\text{E-NFE} = \mathbb{E}\left[\sum_{i=1}^N \mathbb{I}_{\boldsymbol{x}_{t_{i-1}} \neq \boldsymbol{x}_{t_i}}\right] = \sum_{i=1}^N \mathbb{E}\left[\mathbb{I}_{\boldsymbol{x}_{t_{i-1}} \neq \boldsymbol{x}_{t_i}}\right] = \sum_{i=1}^N P\left(\boldsymbol{x}_{t_{i-1}} \neq \boldsymbol{x}_{t_i}\right)$$
$$= \sum_{i=1}^N \left(1 - P\left(\boldsymbol{x}_{t_{i-1}} = \boldsymbol{x}_{t_i}\right)\right) = \sum_{i=1}^N \left(1 - P\left(x_{t_{i-1}}^{(l)} = x_{t_i}^{(l)}, l = 1, 2, \ldots, BL\right)\right) \tag{71}$$

As noted in the main text, an unmasked token will no longer change, and whether a mask token will change during a sampling step is independent of the network output. Given that the reverse sampling process is factorized across dimensions, and the only interaction between dimensions is through the sequence-conditioned network, which does not affect whether a token will change, we conclude that the events $\{x_{t_{i-1}}^{(l)} = x_{t_i}^{(l)}\}$ are independent for different $l$. Therefore, the probability $P\left(x_{t_{i-1}}^{(l)} = x_{t_i}^{(l)}, l = 1, 2, \ldots, BL\right)$ can be factorized as $\prod_{l=1}^{BL} P\left(x_{t_{i-1}}^{(l)} = x_{t_i}^{(l)}\right)$, where

$$P\left(x_{t_{i-1}}^{(l)} = x_{t_i}^{(l)}\right) = P\left(x_{t_i}^{(l)} = m\right) P\left(x_{t_{i-1}}^{(l)} = m | x_{t_i}^{(l)} = m\right) + P\left(x_{t_i}^{(l)} \neq m\right)$$
$$= \frac{1-\alpha_{t_i}}{1-\alpha_1}\frac{1-\alpha_{t_{i-1}}}{1-\alpha_{t_i}} + 1 - \frac{1-\alpha_{t_i}}{1-\alpha_1} \tag{72}$$
$$= 1 - (\alpha_{t_{i-1}} - \alpha_{t_i})$$

The expected NFE is finally simplified to

$$\text{E-NFE} = \sum_{i=1}^N \left(1 - \left(1 - (\alpha_{t_{i-1}} - \alpha_{t_i})\right)^{BL}\right) \tag{73}$$

Using the default linear noise schedule $\alpha_t = 1 - t$ as well as uniform timesteps $t_k = \frac{k}{N}$, we have $\alpha_{t_{i-1}} - \alpha_{t_i} = \frac{1}{N}$, and the expected NFE is $N\left(1 - (1 - \frac{1}{N})^{BL}\right)$.

# F    A BRIEF INTRODUCTION TO GUMBEL TRICKS

Gumbel tricks are widely used in machine learning and statistics to handle the challenges associated with discrete random variables. Based on the properties of the Gumbel distribution, these techniques offer powerful tools for approximating discrete distributions and optimizing over discrete spaces, facilitating the integration of discrete variables into continuous models.

### F.1 THE GUMBEL DISTRIBUTION

The Gumbel distribution (Gumbel, 1935) is related to the extreme value theory and is commonly used to model the distribution of the maximum of random variables. The Gumbel distribution $\mathcal{G}(\mu, \beta)$, with the location parameter $\mu$ and the scale parameter $\beta$, has the following PDF and CDF:

$$F(x; \mu, \beta) = e^{-e^{-(x-\mu)/\beta}}, \quad f(x; \mu, \beta) = \frac{1}{\beta} e^{-(x-\mu)/\beta - e^{-(x-\mu)/\beta}} \tag{74}$$

A widely used special case is the standard Gumbel distribution $\mathcal{G}(0, 1)$, where $\mu = 0$ and $\beta = 1$. For this distribution, the CDF is given by $P(g \leq x) = e^{-e^{-x}}$. Using inverse transform sampling, a random variable $g$ following $\mathcal{G}(0, 1)$ can be sampled by drawing $u \sim \mathcal{U}(0, 1)$ and performing two negative logarithm operations $g = -\log(-\log u)$.

### F.2 GUMBEL-MAX TRICK

The Gumbel-max trick (Gumbel, 1954) allows us to sample from a categorical distribution using continuous random variables. It leverages the following property of the Gumbel distribution: for $\boldsymbol{\pi} = (\pi_1, \pi_2, \ldots, \pi_K)$ satisfying $\boldsymbol{\pi} \geq 0$ and $\sum_{i=1}^{K} \pi_i > 0$, and let $g_1, \ldots, g_K$ be independent samples from $\mathcal{G}(0, 1)$, we have

$$\max_i (g_i + \log \pi_i) \sim \mathcal{G}\left(\log \sum_{i=1}^{K} \pi_i, 1\right) \tag{75}$$

and

$$\underset{i}{\operatorname{argmax}} (g_i + \log \pi_i) \sim \operatorname{Cat}\left(\frac{\boldsymbol{\pi}}{\sum_{i=1}^{K} \pi_i}\right) \tag{76}$$

This implies that sampling a discrete variable from a categorical distribution can be achieved by operating on continuous variables that include the known class probabilities and the sampled Gumbel variables. Therefore, the Gumbel-max trick acts as the *reparameterization trick* for categorical sampling, akin to the Gaussian case used in variational auto-encoders (Kingma & Welling, 2014).

### F.3 GUMBEL-SOFTMAX DISTRIBUTION

The Gumbel-Softmax distribution (Jang et al., 2016) introduces a differentiable approximation to the categorical distribution, facilitating gradient-based optimization in neural network training. It smooths the non-differentiable `argmax` operation in the Gumbel-max trick by replacing it with the differentiable `Softmax` function plus a temperature factor $T$. Specifically, the one-hot random vector $\boldsymbol{e}_x$, where $x \sim \operatorname{Cat}(\boldsymbol{\pi})$, is approximated by a continuous vector $\boldsymbol{y}$ defined as

$$y_i = \frac{e^{(\log \pi_i + g_i)/T}}{\sum_{j=1}^{K} e^{(\log \pi_j + g_j)/T}}, \quad i = 1, 2, \ldots, K \tag{77}$$

It approaches the one-hot representation of the categorical variable when $T \to 0$, and tends to be uniform when $T \to \infty$.

## G IMPLEMENTATION DETAILS

### G.1 ALGORITHMS

For parallel decoding, suppose the sampling step is $N$ and the sequence length is $L$, we define a decoding schedule $\{L_n\}_{n=1}^{N}$ which satisfies $\sum_{n=1}^{N} L_n = L$ to specify the number of tokens decoded at each step. This includes the token-by-token decoding as a special case where $N = L$ and $L_n = 1$. In practice, we decode the same number of tokens per step so that $L$ is divisible by $N$.

We present the parallel decoding procedure in Algorithm 2, which can be interpreted as a first-order method. Algorithm 3 and Algorithm 4 describe two types of high-order extensions, inspired by high-order numerical differential equation solvers in diffusion models. Algorithm 3 leverages

Lagrange polynomials to interpolate the previous network outputs along the time axis, yielding an approximate network prediction for the current time step. Our implementation only uses the two most recent predictions, making it a second-order method, as we empirically find that higher-order methods tend to degrade performance. Algorithm 4 employs a predictor-corrector approach, refining the first-order decoding result at the last step using the current network prediction, also resulting in a second-order method. After refining the intermediate sample, we avoid feeding it back into the network for prediction updates, thus preventing extra NFEs.

---

**Algorithm 2** First-Hitting Sampling of MDMs (parallel decoding)

---

**Require:** the sequence length $L$, the vocabulary $\mathcal{X} = \{0, \dots, m-1, m\}$ where $m$ is the mask token, the noise schedule $\alpha_t$ and its inverse function $\alpha^{-1}$, the pretrained masked diffusion model $\boldsymbol{\mu}_\theta$, the number of sampling steps $N$, the decoding schedule $\{L_n\}_{n=1}^N$

 1: $\boldsymbol{x}_L \leftarrow [m\ m\ \dots\ m]$
 2: $\tau_L \leftarrow 1$
 3: $l \leftarrow L$
 4: **for** $n \leftarrow N$ **to** 1 **do**
 5:     **for** $i \leftarrow 1$ **to** $L_n$ **do**
 6:         Sample $u_l \sim \mathcal{U}(0,1)$
 7:         $\tau_{l-1} \leftarrow \alpha^{-1}(1 - u_l^{1/l}(1 - \alpha_{\tau_l}))$
 8:         **if** $i = 1$ **then**
 9:             $\boldsymbol{\mu} \leftarrow \boldsymbol{\mu}_\theta(\boldsymbol{x}_l, \tau_{l-1})$
10:         **end if**
11:         Randomly and uniformly select an index $k$ from $\{j : x_l^{(j)} = m\}$ (i.e., masked positions in $\boldsymbol{x}_l$)
12:         $\boldsymbol{x}_{l-1} \leftarrow \boldsymbol{x}_l, x_{l-1}^{(k)} \leftarrow x \sim \mathrm{Cat}(\boldsymbol{\mu}^{(k)})$
13:         $l \leftarrow l - 1$
14:     **end for**
15: **end for**
**Output:** $\boldsymbol{x}_0$

---

---

**Algorithm 3** First-Hitting Sampling of MDMs (extrapolation)

---

**Require:** the sequence length $L$, the vocabulary $\mathcal{X} = \{0, \dots, m-1, m\}$ where $m$ is the mask token, the noise schedule $\alpha_t$ and its inverse function $\alpha^{-1}$, the pretrained masked diffusion model $\boldsymbol{\mu}_\theta$, the number of sampling steps $N$, the decoding schedule $\{L_n\}_{n=1}^N$

 1: $\boldsymbol{x}_L \leftarrow [m\ m\ \dots\ m]$
 2: $\tau_L \leftarrow 1$
 3: $l \leftarrow L$
 4: **for** $n \leftarrow N$ **to** 1 **do**
 5:     **for** $i \leftarrow 1$ **to** $L_n$ **do**
 6:         Sample $u_l \sim \mathcal{U}(0,1)$
 7:         $\tau_{l-1} \leftarrow \alpha^{-1}(1 - u_l^{1/l}(1 - \alpha_{\tau_l}))$
 8:         **if** $i = 1$ **then**
 9:             $\boldsymbol{\mu} \leftarrow \boldsymbol{\mu}_\theta(\boldsymbol{x}_l, \tau_{l-1})$
10:             $\tau \leftarrow \tau_{l-1}$
11:         **end if**
12:         **if** $n = N$ **then**
13:             $\hat{\boldsymbol{\mu}} = \boldsymbol{\mu}$
14:         **else**
15:             $\hat{\boldsymbol{\mu}} = \dfrac{\tau_{l-1} - \tilde{\tau}}{\tau - \tilde{\tau}}\boldsymbol{\mu} + \dfrac{\tau_{l-1} - \tau}{\tilde{\tau} - \tau}\tilde{\boldsymbol{\mu}}$ (Lagrange interpolation)
16:         **end if**
17:         Randomly and uniformly select an index $k$ from $\{j : x_l^{(j)} = m\}$ (i.e., masked positions in $\boldsymbol{x}_l$)
18:         $\boldsymbol{x}_{l-1} \leftarrow \boldsymbol{x}_l, x_{l-1}^{(k)} \leftarrow x \sim \mathrm{Cat}(\hat{\boldsymbol{\mu}}^{(k)})$
19:         $l \leftarrow l - 1$
20:     **end for**
21:     $\tilde{\boldsymbol{\mu}} \leftarrow \boldsymbol{\mu}$
22:     $\tilde{\tau} \leftarrow \tau$
23: **end for**
**Output:** $\boldsymbol{x}_0$

---

---

**Algorithm 4** First-Hitting Sampling of MDMs (predictor-corrector)

---

**Require:** the sequence length $L$, the vocabulary $\mathcal{X} = \{0, \ldots, m-1, m\}$ where $m$ is the mask token, the noise schedule $\alpha_t$ and its inverse function $\alpha^{-1}$, the pretrained masked diffusion model $\boldsymbol{\mu}_\theta$, the number of sampling steps $N$, the decoding schedule $\{L_n\}_{n=1}^N$

1: $\boldsymbol{x}_L \leftarrow [m\ m\ \ldots\ m]$
2: $\tau_L \leftarrow 1$
3: $l \leftarrow L$
4: **for** $n \leftarrow N$ **to** $1$ **do**
5:   **for** $i \leftarrow 1$ **to** $L_n$ **do**
6:    Sample $u_l \sim \mathcal{U}(0, 1)$
7:    $\tau_{l-1} \leftarrow \alpha^{-1}(1 - u_l^{1/l}(1 - \alpha_{\tau_l}))$
8:    **if** $i = 1$ **then**
9:     $\boldsymbol{\mu} \leftarrow \boldsymbol{\mu}_\theta(\boldsymbol{x}_l, \tau_{l-1})$
10:     **if** $n < N$ **then**
11:      $\boldsymbol{x}_l \leftarrow \hat{\boldsymbol{x}}$
12:      **for** $r \leftarrow 1$ **to** $L_{n+1}$ **do**
13:       Randomly and uniformly select an index $k$ from $\{j : x_l^{(j)} = m\}$
14:       $x_l^{(k)} \leftarrow x \sim \text{Cat}(\boldsymbol{\mu}^{(k)})$
15:      **end for**
16:     **end if**
17:     $\hat{\boldsymbol{x}} \leftarrow \boldsymbol{x}_l$
18:    **end if**
19:    Randomly and uniformly select an index $k$ from $\{j : x_l^{(j)} = m\}$ (i.e., masked positions in $\boldsymbol{x}_l$)
20:    $\boldsymbol{x}_{l-1} \leftarrow \boldsymbol{x}_l, x_{l-1}^{(k)} \leftarrow x \sim \text{Cat}(\boldsymbol{\mu}^{(k)})$
21:    $l \leftarrow l - 1$
22:   **end for**
23: **end for**
**Output:** $\boldsymbol{x}_0$

---

### G.2 Low-Discrepancy Sampler

VDM (Kingma et al., 2021) proposes a low-discrepancy sampler for batched sampling of uniformly distributed continuous time variables, which reduces the loss variance in maximum likelihood training of diffusion models. Specifically, consider a batch of $B$ timesteps $t^{(i)}{}_{i=0}^{B-1}$ that needs to be sampled from $\mathcal{U}(0, 1)$. Instead of sampling them independently, VDM generates correlated samples using the formula $t^{(i)} = \text{mod}(u_0 + i/B, 1)$, where $u_0 \sim \mathcal{U}(0, 1)$. This approach ensures that each $t^{(i)}$ has the correct marginal distribution over multiple batches, while each batch of timesteps more evenly covers the interval $[0, 1]$.

MDLM (Sahoo et al., 2024) employs a slightly different low-discrepancy sampler where the sampled timesteps are less correlated within a batch. Specifically, $B$ independent uniform samples $\{u_i\}_{i=0}^{B-1}$ from $\mathcal{U}(0, 1)$ are mapped into $B$ bins by $t^{(i)} = (u_i + i)/B$. We adopt this approach for sampling continuous timesteps. Additionally, we extend this low-discrepancy sampler to handle discrete timesteps $\{n^{(i)}\}_{i=0}^{B-1}$ drawn from $\mathcal{U}(\{0, 1, \ldots, L-1\})$ (e.g., the number of masked tokens in the discrete ELBO) by mapping the continuous time $t^{(i)}$ to $n^{(i)} = \lceil Lt^{(i)} \rceil$.

## H  Experiment Details

### H.1 Evaluation Metrics

**Perplexity** is a likelihood-related metric to evaluate how well a likelihood-based model is trained. Denote the likelihood (i.e., the probability of the data under the parameterized model) for the data

point $\boldsymbol{x}_0$ as $p_\theta(\boldsymbol{x}_0)$. The log-likelihood can be expressed either exactly or through an ELBO:

$$\text{Auto-regressive Models:} \quad \log p_\theta(\boldsymbol{x}_0) = \sum_{n=1}^{L} \log p_\theta(x_0^{(n)}|\boldsymbol{x}_0^{(<n)}) \tag{78}$$

$$\text{Masked Models:} \quad \log p_\theta(\boldsymbol{x}_0) \geq \sum_{n=1}^{L} \mathbb{E}_{\tilde{q}_{n|0}(\boldsymbol{x}_n|\boldsymbol{x}_0)} \left[ \frac{1}{n} \sum_{l:x_n^{(l)}=m} \log p_\theta(x_0^{(l)}|\boldsymbol{x}_n) \right] \tag{79}$$

Here we express the log-likelihood of masked models with our derived discrete ELBO. We use $\log p_\theta(x_0^{(l)}|\boldsymbol{x}_n)$ (predicted data probability given known tokens) as an alternative expression for the cross-entropy term $\boldsymbol{e}_{x_0^{(l)}}^\top \log \boldsymbol{\mu}_\theta^{(l)}(\boldsymbol{x}_n)$, aligning it with the common formulation in ARMs. The perplexity (PPL) is defined as:

$$\text{PPL} = \exp\left( \frac{\mathbb{E}_{\boldsymbol{x}_0 \sim p_{\text{data}}}[-\log p_\theta(\boldsymbol{x}_0)]}{D} \right) \tag{80}$$

where $D$ is the data dimension, and $p_{\text{data}}$ is the data distribution (such as the validation/test set).

**Generative Perplexity** evaluates a model's generation quality by measuring the perplexity of its generated samples under some off-the-shelf model. It is related to both training and sampling. We adopt GPT-2 Large as the off-the-shelf evaluator following previous works.

**Entropy** measures the diversity of tokens in a sequence. For a sequence of length $L$ that contains $K$ distinct tokens, with each token $k$ occurring $L_k$ times, the entropy is computed as $-\sum_{k=1}^{K} p_k \log p_k$, where $p_k = L_k/L$ represents the probability of occurrence of token $k$.

### H.2 MODEL AND DATASET DETAILS

Following SEDD (Lou et al., 2023) and MDLM (Sahoo et al., 2024), we utilize an encoder-only transformer with a DDiT (Peebles & Xie, 2023) architecture, incorporating RoPE (Su et al., 2024). We use the small-size model variant, which consists of 12 layers, 12 attention heads, a hidden dimension of 768, and a timestep embedding dimension of 128, amounting to approximately 170M parameters including the word embedding matrix.

Our experiments are conducted on the OpenWebText dataset (Gokaslan et al., 2019), which contains around 8 million documents, with the last 100k reserved for validation. The dataset is tokenized using the GPT-2 tokenizer, resulting in a vocabulary size of 50,257 (excluding the mask token). Sequences are concatenated and wrapped to a length of 1024 tokens, with the first, last, and in-between tokens of concatenated sequences set to `eos`.

### H.3 TRAINING DETAILS

Following SEDD (Lou et al., 2023) and MDLM (Sahoo et al., 2024), we use the AdamW optimizer with a batch size of 512 and a learning rate that is linearly warmed up from 0 to 3e-4 over the first 2,500 steps. We apply a dropout rate of 0.1, clip the gradient norm to 1, and utilize an exponential moving average (EMA) with a rate of 0.9999. Mixed-precision training is enabled with `bfloat16`.

All our training experiments are conducted on 8 NVIDIA A100 40GB GPUs for slightly over 100k iterations, which takes around 1.5 days.

### H.4 SAMPLING DETAILS

We directly use the pretrained models (AR, SEDD Absorb, MDLM) trained on OpenWebText provided by MDLM[11]. These models share the same architecture and size (with the exception of the final layer in AR). SEDD and MDLM are trained for 1M iterations, while the corresponding AR baseline is trained for half as many steps to ensure a comparable number of tokens seen.

For the baselines, SEDD is sampled using its analytic sampler (Tweedie $\tau$-leaping), and MDLM is sampled both with and without the caching strategy. Their sampling timesteps are uniformly discretized. In our first-hitting sampler, parallel decoding is achieved by unmasking the same number

---

[11]https://github.com/kuleshov-group/mdlm/

of tokens at each step. Although the pretrained MDLM model is claimed to be time-independent, we find that adding the time condition slightly improves performance in our sampler. All sampling experiments are conducted on a single NVIDIA RTX A6000 GPU, and the reported metrics are averaged on 64 random samples.

## I  OLD VERSION OF THE INTRODUCTION

We ***highlight*** our key findings: (1) MDMs, in both training and sampling, are essentially time-agnostic masked models (or order-agnostic auto-regressive models), enjoying **20×** faster sampling and diverging from the design choices of diffusion models. This also justifies the theoretical foundation of masked models as they are equivalent and simpler formulations of the more principled MDMs. (2) We challenge previous claims that MDMs can surpass ARMs in text generation by identifying a hidden but critical numerical issue that reduces the token diversity and renders previous evaluations unfair. After fixing it, we find MDMs significantly lagging behind ARMs in generative perplexity.

For training, we prove that the continuous-time evidence lower bound (ELBO) objective of MDMs can be expressed by the number of masked tokens with an implicitly defined mixture-of-experts model. It provides a discrete ELBO for masked models and coincides with the ELBO previously derived for order-agnostic auto-regressive models (Uria et al., 2014; Hoogeboom et al., 2021a).

For sampling, by analytically sampling the time when any mask token is first unmasked, we propose a theoretically equivalent first-hitting sampler (FHS) to avoid most of the time-consuming categorical sampling and perform decoding token by token with no approximation errors. It is further extended to enable parallel decoding and incorporate high-order approximations, achieving a 20× speedup compared to previous MDM sampling procedures. When the parameterized model is independent of the time variable, we recover the sampling of masked models.

For evaluation, we discover that while MDMs exhibit extremely low generative perplexity with numerous sampling steps, the generation quality is compromised by reduced token diversity. By examining the numerical precision during sampling, we identify a previously unrecognized issue with Gumbel-based categorical sampling. Specifically, reducing the floating-point precision from 64-bit to 32-bit significantly truncates the Gumbel variables, which theoretically lowers the temperature and empirically improves the generative perplexity of pretrained models (Lou et al., 2023; Sahoo et al., 2024) from 126.11 to 31.24, but with a decreased sentence entropy from 5.66 to 5.17.

## J  ADDITIONAL RESULTS

### J.1  TRAINING RESULTS

#### J.1.1  COMPARISON OF TRAINING VARIANTS

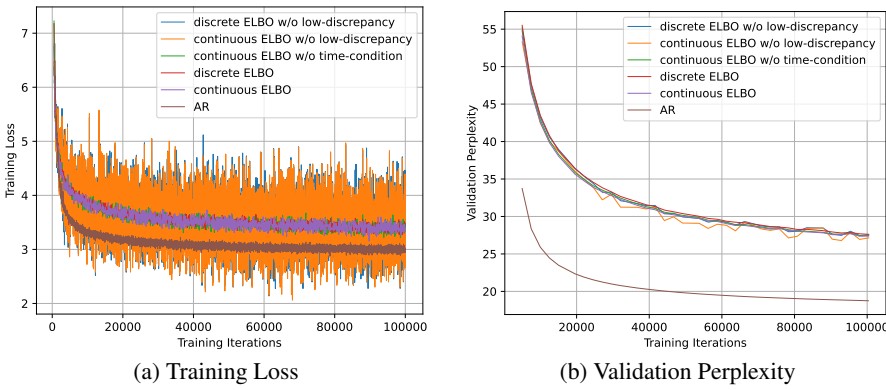

(a) Training Loss  (b) Validation Perplexity

Figure 11: Comparison of training variants.

We compare different training variants (continuous-time/discrete ELBO, time-conditioned/time-independent network) of MDMs. By default, we condition the network on time for continuous-time ELBO and on the masked ratio for discrete ELBO. We also apply the low-discrepancy sampler described in Appendix G.2. Providing the network with extra conditions as auxiliary information may potentially facilitate the training process.

As shown in Figure 11, all variants exhibit similar performance in both training and validation. Adding the time condition provides a slight improvement over the time-independent network, and the discrete ELBO performs marginally worse than the continuous-time ELBO. However, these differences are negligible. The low-discrepancy sampler notably reduces the variance in training loss, though the validation perplexity curve remains relatively stable, likely due to the smoothing effect of the EMA. Nonetheless, MDMs still significantly lag behind the counterpart ARM.

### J.1.2 FAILED TRAINING ATTEMPTS

Training MDMs with the ELBO is analogous to maximum likelihood training of diffusion models (Song et al., 2021b; Kingma et al., 2021; Lu et al., 2022a; Zheng et al., 2023b). Therefore, we borrow well-established techniques from the SOTA likelihood model i-DODE (Zheng et al., 2023b) within the diffusion literature, including velocity parameterization and variance reduction.

**Flow Matching/Preconditioning** Different parameterizations are theoretically equivalent but have distinct empirical implications in diffusion models. As an alternative to data or noise prediction, velocity parameterization has proven effective in the maximum likelihood training of diffusion models (Zheng et al., 2023b). It is also related to flow matching (Lipman et al., 2022) and the preconditioning technique in EDM (Karras et al., 2022). As MDMs employ mean parameterization (or data prediction), exploring alternative parameterizations may enhance training performance.

In diffusion models, different parameterizations can be understood as expressing the mean prediction model $\boldsymbol{\mu}_\theta$ with a "skip-connection" style preconditioning:

$$\boldsymbol{\mu}_\theta(\boldsymbol{x}_t, t) = a_t \boldsymbol{F}_\theta(\boldsymbol{x}_t, t) + b_t \boldsymbol{x}_t \tag{81}$$

where $a_t, b_t$ are some specific time-related coefficients, and $\boldsymbol{F}_\theta$ is a free-form network. In MDMs, this general preconditioning can be formulated by the one-hot vector $\boldsymbol{e}_{x_t^{(l)}}$:

$$\boldsymbol{\mu}_\theta^{(l)}(\boldsymbol{x}_t, t) = a_t \boldsymbol{F}_\theta^{(l)}(\boldsymbol{x}_t, t) + b_t \boldsymbol{e}_{x_t^{(l)}} \tag{82}$$

However, such a formulation in MDMs makes no difference to the training. $\boldsymbol{\mu}_\theta^{(l)}(\boldsymbol{x}_t, t)$ is only trained when $x_t^{(l)} = m$ to predict the data probabilities at dimensions $0 \sim m-1$. Adding $\boldsymbol{e}_{x_t^{(l)}}$ (which is 0 at dimension $0 \sim m-1$) to the model output does not impact the functional dimensions.

To tackle this, we attempt to employ a self-conditioning technique (Chen et al., 2022)

$$\boldsymbol{\mu}_\theta(\boldsymbol{x}_t, t) \coloneqq \boldsymbol{F}_\theta(\boldsymbol{x}_t, t, \boldsymbol{F}_{\theta^-}(\boldsymbol{x}_t, t)) \tag{83}$$

where $\theta^-$ is the stop-gradient version of $\theta$. For implementation, the extra condition $\boldsymbol{F}_{\theta^-}$ (1) is concatenated to the original input along the feature dimension (2) is replaced by blank with 50% probability during training, and substituted by the earlier model prediction during inference. However, we empirically find this technique highly unstable during training, adding extra model parameters and incurring excessive training costs.

**Variance Reduction via Importance Sampling** The NELBO for both diffusion models and MDMs can be expressed as an expectation $\mathcal{L} = \mathbb{E}_t[\mathcal{L}_t]$ over uniformly distributed $t$, where $t$ can be either continuous (e.g., the continuous time) or discrete (e.g., the discrete time or the number of masked tokens). Importance sampling (IS) for $t$ can be introduced by rewriting the training loss with a proposal distribution $p_t$:

$$\mathcal{L} = \mathbb{E}_{t \sim p_t}\left[\tilde{\mathcal{L}}_t\right], \quad \tilde{\mathcal{L}}_t = \frac{\mathcal{L}_t}{p_t} \tag{84}$$

While the overall loss $\mathcal{L}$ remains invariant to the choice of $p_t$, the variance of the loss, $\mathrm{Var}_{t \sim p_t}\left[\tilde{\mathcal{L}}_t\right]$, is influenced by $p_t$. We can optimize $p_t$ for variance minimization. For continuous $t$, the density

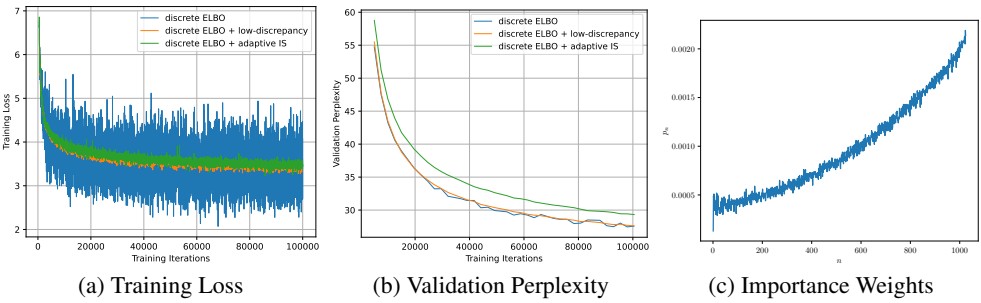

(a) Training Loss      (b) Validation Perplexity      (c) Importance Weights

Figure 12: Variance reduction methods.

$p_t$ can be parameterized by a monotonic neural network and learned by gradient descent (Kingma et al., 2021; Zheng et al., 2023b).

For simplicity, we instead consider the discrete case of $t$ (the number of masked tokens in the discrete ELBO). We adopt the adaptive IS proposed by Improved DDPM (Nichol & Dhariwal, 2021). Specifically, as $\mathrm{Var}_{t \sim p_t}\left[\tilde{\mathcal{L}}_t\right] = \mathbb{E}_{t \sim p_t}\left[\tilde{\mathcal{L}}_t^2\right] - \mathcal{L}^2$, the optimal $p_t$ is given by $p_t \propto \sqrt{\mathbb{E}\left[\mathcal{L}_t^2\right]}$. Given that $\mathbb{E}\left[\mathcal{L}_t^2\right]$ is unknown in advance and may vary throughout training, we maintain a history of the previous 10 values for each loss term and update this dynamically during training. At the beginning of training, we sample $t$ uniformly until we have 10 samples for every $t$.

We visualize the training curves and optimized importance weights at around 100k iterations in Figure 12. Unfortunately, while the adaptive IS technique reduces the variance to a level comparable to the low-discrepancy sampler, it also results in degraded performance. We hypothesize that the dynamical updated $p_t$ may make the loss estimator biased.

## J.2    SAMPLING RESULTS

### J.2.1    COMPARISON OF HIGH-ORDER VARIANTS

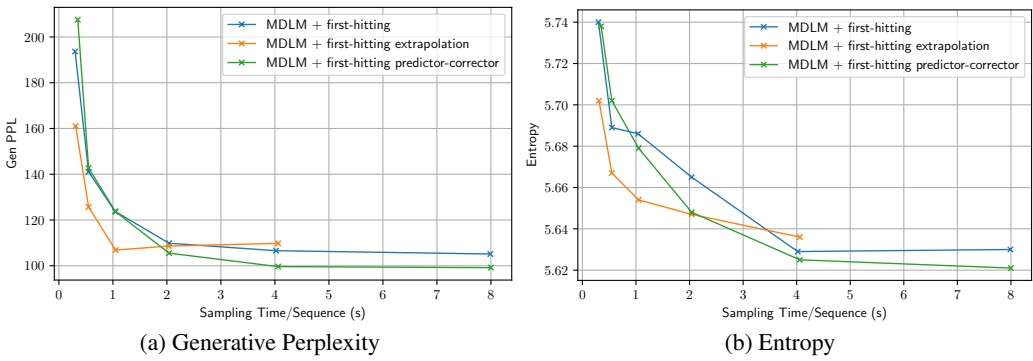

(a) Generative Perplexity      (b) Entropy

Figure 13: Comparisons of high-order variants of the first-hitting sampler with steps $N \in \{32, 64, 128, 256, 512, 1024\}$.

In Figure 13, we compare the two high-order variants of our proposed first-hitting sampler. In terms of the generative perplexity, The extrapolation strategy performs best when $N \leq 128$, and the predictor-corrector strategy is more effective when $N \geq 256$. We also observe that lower generative perplexity is associated with decreased entropy, indicating an inherent trade-off.

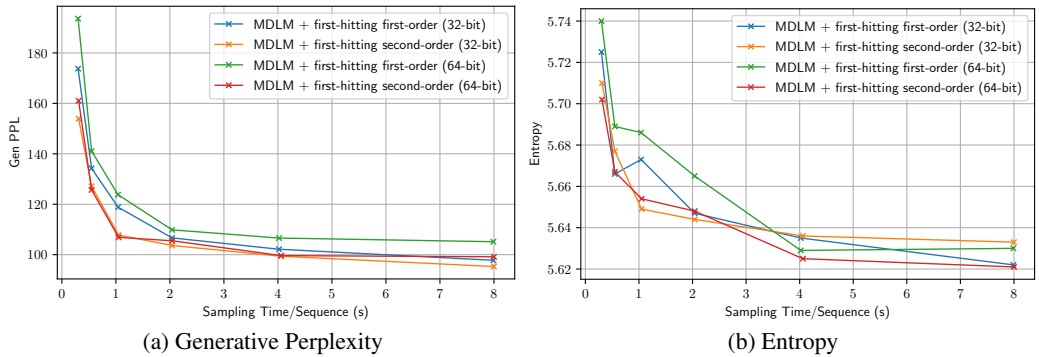

(a) Generative Perplexity

(b) Entropy

Figure 14: Impact of numerical precision on the first-hitting sampler with steps $N \in \{32, 64, 128, 256, 512, 1024\}$.

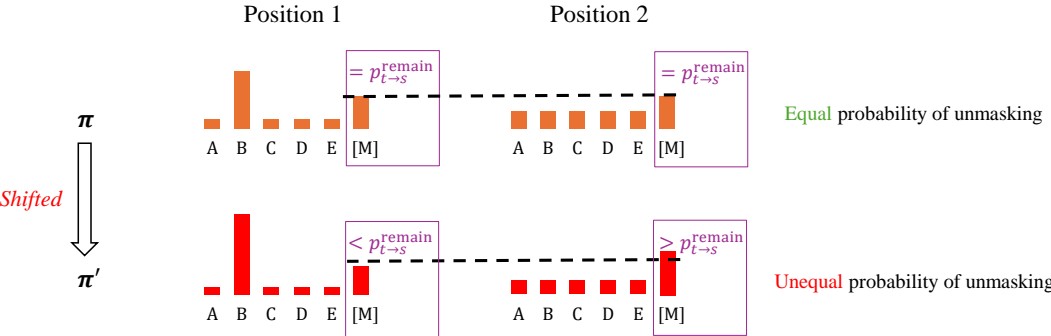

Figure 15: Illustration of prioritized unmasking.

### J.2.2 IMPACT OF NUMERICAL PRECISION ON OUR SAMPLER

In Figure 14, we examine the impact of numerical precision on our first-hitting sampler by varying the floating-point precision in categorical sampling between 32-bit and 64-bit. For the high-order variants, we employ the extrapolation strategy when $N \leq 128$ and the predictor-corrector strategy when $N \geq 256$. In contrast to the observations under MDM's original sampler, the temperature-lowering effect of the numerical truncation is significantly less influential under our sampler. Notably, the 32-bit second-order first-hitting sampler even results in slightly higher entropy compared to its 64-bit counterpart when $N \geq 512$.

**Why the numerical issue is not notable in token-by-token decoding**   With our first-hitting sampler, the inference of MDMs becomes a token-by-token decoding process, except that the time variable is additionally handled. In this case, the numerical issue becomes negligible, and 32-bit floating-point precision appears sufficient. We also observe that ARMs, which also adopt a token-by-token sampling strategy, do not suffer from numerical issues under 32-bit precision. This suggests that *the numerical problem is a distinctive characteristic of the vanilla sampling process of MDMs (Eqn. (10)) that performs inaccurate categorical sampling simultaneously on all mask positions.*

This phenomenon can be explained as follows. As justified theoretically in Section 5.3, the implication of inaccurate categorical sampling includes two aspects: (1) temperature-lowering effect for a single token position and (2) prioritized unmasking for different positions (Figure 15). Both factors reduce the diversity and lower the entropy. However, when altering to token-by-token decoding, all remaining mask tokens have equal probability to be first unmasked, and the diversity decrease becomes less pronounced. This suggests that *the prioritized unmasking caused by shifted probabilities at all individual positions is the major factor.* This effect accumulates across numerous sampling steps, eventually leading to notable diversity issues, even under 32-bit floating-point precision.

## J.3 Anaysis of the Efficiency Gain by Our Sampler

Due to the theoretical equivalence between the FHS and the original MDM sampling procedure, the correctness of the FHS is guaranteed and irrelevant to vocabulary size or sequence length. However, the efficiency gains (measured by inference wall-clock time) depend on several factors.

Let the sequence length be denoted as $L$, vocabulary size as $|V|$ (excluding the mask token), the number of sampling steps (for original MDM sampling) as $N$, the number of function evaluations as $NFE$, and the number of categorical sampling operations as $NCS$. As stated in Section 4, our FHS reduces inference time by minimizing $NCS$. For original MDM sampling with the caching strategy, $NFE \approx N(1 - (1 - 1/N)^L)$, $NCS = NL|V|$. For the FHS, $NCS = L|V|$. The total time cost can be expressed as $NFE \times t_1 + NCS \times t_2$, where

- $t_1$ is the time for one network call, influenced by model size.
- $t_2$ is the time for categorical sampling, averaged per position and class, which involves two logarithmic operations, and is fixed.

For a fair comparison, we evaluate under the **same** $NFE$, ensuring similar generation quality. The inference time ratio between original MDM sampling and the FHS is $(NFE \times t_1 + NL|V|t_2) : (NFE \times t_1 + L|V|t_2)$. Therefore, the FHS yields larger speedups under the following conditions:

- **Smaller Model Size:** When the model is smaller, $t_1$ decreases, making categorical sampling relatively more expensive compared to network evaluation.
- **Larger $NFE$, $L$, or $|V|$:** Higher values for these parameters increase $NCS$ in original MDM sampling, amplifying the speedup provided by the FHS.

Examples:

- **Paper Case:** $|V| = 50{,}526$, $L = 1024$. To match the original MDM sampling at $N = 10{,}000$ steps, $NFE \approx 973$. The model size is approximately 600M parameters, with a time ratio of $t_1 : L|V|t_2 \approx 1 : 1.56$. In this case, the inference time ratio is around 17, corresponding to a **20×** **speedup**. If $N = 2048$, the ratio drops to around **5×**.
- **DiffSound** (Yang et al., 2023): This model is 3× larger and uses a much smaller vocabulary. Specifically, $L = 265$, $|V| = 256$, with $t_1 : L|V|t_2 \approx 14.6 : 1$. Here, categorical sampling is much cheaper relative to network evaluations. Using fewer steps, $N = 100$, results in $NFE \approx 93$, and the speedup ratio is only around **1.07×**.

## J.3.1 Examples of Generated Text

<|endoftext|> the new cars are crossovers.

AT&T Insurance Marketing Manager, Megan Maxwell, tells us that Model X was "reasonably priced, effective and inspires strong sentiment among consumers." She says:

Our GM car for discussion is shown as part of our drive 20 percent around the world and even a competitor. Our GM for discussion alt shows as one of our most popular cars in the world. We are in multiple countries introducing firmware for our new vehicles. While we are confident in our prices, we rely upon GM Auto's sales data and know we must adapt this process to meet the needs of all customers.

The proposed pricing is similar to that of the cheaper Range Rover and other cheaper sport utility vehicles, which are primarily offered through its dealerships. Alongside a Volt, Delphi XE8 includes a plug-in hybrid version called Volt Energy.

"Dynamic pricing is our way to deliver owners of more attractive or more reasonable outcomes or to find more marketable models that appeal to them more than their competitors," notes Maxwell.

Earlier this week, GM analyst Greg Clifford predicted that Intel Global Radical Charge Power Savings (STB) would start at \$3,300 over the product lifecycle with an adoption rate of 50 percent by 2025.<|endoftext|>The Warner Bros. foreign distribution arm The Weinstein Company tried to keep the Weinstein Company character Harvey Weinstein out of The Bourne Identity, but now that the character has been ousted the distributor has effectively banned him from the Middle East or North Africa.

Caici International Union of Arabic and Al-Ahly Travel Negotiations president Shadi Hamid tweeted Thursday:

A merciful Allah lifts my people wherever they are, provided that I am safe and secure. — Shadi Hamid (@shadihamid) June 26, 2014

An anonymous official familiar with the matter tells The Hollywood Reporter that the tentative suspension of the sale was prompted by a request by Weinstein's Company that The Weinstein Company not use the #BourneCostabool hashtag or its photo that appeared in the film. Weinstein's studio says he hasn't spoken with the history-making movie star after the film was released on Nov. 19 but that Trump's inauguration had somewhat diminished the company's image abroad. The awards-winning film grossed more than \$411 million globally.

EARLIER:

Original Screenplay for 'The Bourne Identity' Is Now Removed

Grisly Official Appointments Express for Harvey Weinstein Probing

Bourne Identity Alleged Tarantino Power Grab Shot with \$42 Million Spent on Hollywood Update: is just re-issued as The Weinstein Company ban shows nothing.

BREAKING: Mancini exits technicalities in Weinstein's sale of Leadville to Columbia pic dir. exec-to-branch. — Reginald Perag (@ReginaldCAeg) June 26, 2014

Leaked From Studio #BourneCostabool. I came out with it after #HarveyTurnedEmDown. — Reginald Perag (@ReginaldCAeg) June 25, 2014<|endoftext|>NETWORK FINALS: H. REILLARY KREATY & HARDBALL added extra 0.1 to final numbers. UNBELIEVABLE INJURY narrowly lost 0.0 from last night. NEW MIX TAPE (9P) +11.0, CNN NEWSROOM +10.6, LUTHER KING 5 +8.1, SOUTH PARK MIDDLE FIVE +2.4, DJ Khaled +2.2, WES CRAFT +1.6, HOUSE HUNTERS INTL AROUND FOOD +0.4, TEEN YEARS MOM 2015 +0.1, TINY HOUSE HUNTERS INTL +0.1, MLB TONIGHT +0.1, FLCL +0.0, SEUDAS FAMILY -0.3, ANZE +0.1, TEEN TITANS GO! +0.1. 10/24/15 –0.3 1.49 ANNCOUNTERS. BARRIERE Z PACK +0.1, NYXY +0.0, OTHER VIEWERS (-0.2) +1.0, A&E MOVIE FIXER UPPER -0.0

Broadcast Official Nationals Program Ratings Chart

CABLE HIGHLIGHTS: UBS CELEBRATES THE DAY! CLOUD CROSSWITCH was at the top with dramatic 4.121, down 23% over last night (4/27), to #4. Football Night in America was #2 (and barely a night after ending), with TIZARD (2.092, 0.919), MASTERS OF KARMA (2.108, 0.914) and SOUTH

Figure 16: The counterpart ARM.

<|endoftext|> I was striking out, he got it to the third out. I know it wasn't really a catch there, but he didn't consider that at all. I knew what it wasn't about. We scored by one run, so it was a difficult thing for me to go to a "get-out," but being on the board is not an easy fact. I knew it was in play, and went to the third out that it needed to be thrown to. I had nothing to say about it, and it was the only opportunity for me to get, to make an appeal that might not come back, and it was the only time he saw that at that moment. And so we continued.

"So we were sitting in the plus-five-and-plus-three-at-night, because we were up on an inning. We were in pinch-hit early in the ninth. I looked at a guy and was very nervous and confused, so when I looked at this man — I can't remember what he was doing, but to double or dead-se the bases for sure — I said you'll be the first one up, then I'm out of the ballpark. I felt like something had to be wrong with me, and he turned to me and told me, "that's as bad as the rest," just like that — "I bet!" I said calmly. "I bet,'" "Gotta do your best! Do not bet!'" "And when that happens, do you think I can be won by a one or two runs?" He stood there and looked stunned. "You mean that?" "Yes, absolutely." "'Yeah, absolutely." "Well, is that a message to you?'"" And he looked back to me and said, "So we aren't going out?" "Well, yes," said me, "but I believe so I believe."" "I bet," "Yeah, I bet, but when we're on the board, how much time are we gonna lose?" "Absolutely not," "I bet," "all right. I'm not going out. It's me, understand." "I bet, I believe not." "What happened?"" "It happened!" "Did you hear a clue?" "I said, "Oh, no! I-I-I heard that fifth-dinger! Give me the clue!" The players, myself, and the "Man, Man, Man, it's just beyond hell!" murmurs of the players. At the same time, I said, myself, "Young man, I've got to say — I won't screw you right here." I went on, "You can take it. You're not going to lose."" I smiled. "You know what you got to admit to yourself? This happened in baseball. I didn't screw you in baseball, it don't matter, I'll screw you in a way." He took the fifth-dinger and said, "It's the end in baseball, it's the end." "That's correct!" I said. "Yes, you can't win in baseball," I said. "But you're not winning in baseball." He turned to me. "No, really, it's alright whether you're winning or not." — "I'm sure," he said. "Good money!" — I cut off. "You're not going to get this out. You do." Those were a few words. As I were thinking, "What an enterprise."

"What are you in baseball?"

"Ah, and it's a game, not a story and a number. If you believe the most in-the-30s stories are about the when-they-had-to-be-done-as-cardinals-but-recan't-they-get-in story?" I said "suck," and "We did stop listening to the number, and we had to come off with the number." "Exactly," said Mike. <|endoftext|>

Figure 17: MDLM with the original sampler (32-bit), 50k steps.

<|endoftext|>I know all about generators and that is one way to build an array.

def call () :  # calls for (t.length, i) = 0: str += " false ",t = (TController person.list objects) func("call".url("")for a,n = t.length: [T in t.T[n])): # def calls [a,n]: * str = str32(T.setUserstr() for a, t & i in i): str += str28(n = [] t[n=i] # put in a list of objects: t = tree() call [T in T.T[n i] = array(t.length) if t.length: [T[i]=[]): string d = [] t stracket3 = tail.stracket(string e) [modify: True] tarring = tail.filesPDF.print(dir: "./file") (piling: True, takeAttempts: false): if hash == 0: print "failed" call r = slacktrace(): names = t.names. relist(cherying) def up(,x, mtr) : p. stuff array. append(:xltr) # So they should have this method def bits(.[1:2]: return "x;0".bit()) #3

Ok, now we want to create a simple list. Try. It's just junk moves. This array can have all of the objects arguments it packed in:

[5].add( {instance.new = new (instance = (instance.new)) (instance = (instance.new)) (console.join(instance.new)) func R[ 3]

In our example, [1, 2], [2, 3, 4, 5 and 8 . . . (0, 31', 42'']' lists contain more elements than in normal arrays and other complex types. It might be significant to say the list includes each engine constant size if it's not slow, but not if it contains no elements. To summarize, we get a list with 3 possible names and divide that list gradually between #3-4.

Now here's how we'll build an array with synthetic element references:

foo = Foo { 3:3:4 } observe a variable from (func yout int y) -> nil do [apply (number of]) assert false if not true # make references for foo bar x (map (variable x y = let x x as par y x x as par y]) (strip 'uoy) print('foo bar foo')) """.add bar x """.add (print('foo foo')) {any,...} observe a tuple from (func (func xs) -> nil do let.add = let element if (element == tuple[numberOf]: 0): ".add(_) done if (done == 1) return "doDone"} ".add (j2.resolve(foo), '(WhatWhy'.by','Call'.d somewhere?'),) end (waffle(j1, (j2.w2](require::John method 'call)(' of ['Ref()']])))))) } kilobyms aren't practical with these from a memory perspective. And don't ask at the very end where lines are the lowest line when they won't be allocated in memory.

(map? # add integers for Lua Lua? # do arithmetic for Emacs) end Lua? def x() # retrieve a++ list item (func (func list)) -> nil do (for method '_of', t) i a] puts t'm "-L','N"3)) return nil def drop_list #5

Added by clip :

Scratch

Now that you know it's adding elements moves again, a lot of other tricks are used in some Perl code. Common Lisp is the lone voice of the Emacs-Stuff. I should be careful just that Lisp is good and can help out, it is easier to make the code readable even than it is to express the tested is really pretty. Having a lot of clause numbers is good enough.

Still, you'll find difficulties in extracting and adding the references pretty amazing. You have to look at code to get it to work. Each line has to introduce a name for each variable and its arguments. Combine the arguments with as1,2,2,3,4,5 . . . , changing this to one which those argument arguments is called. When we are functional it's correct to pass in multiple elements from multiple lists:

[] (x, d) list.remove() d1 . . . (0, 3, (10, 6, 12) (list(or, r))))function () is identical to the line<|endoftext|>

Figure 18: MDLM with the original sampler (64-bit), 50k steps.

<|endoftext|> tough cases be property/private fields in Base class for example Class or with struct containing member.

-Loose the logic to possible the entry point to a species tree:

class Terence___ = 'sum___' collections.create_species_tree( Age = Age(='ab'). # Adult; Reward = Class Agg::PopulationId( ('Founded by:'), - 'Age';Biometric','Nominal','Possible inhabit:', Population','Life') )

To construct a family tree then it uses join(members) creating a chain(MONS in a family tree)

Using ((each ( urn___) multi-found 10..12 sequence heads { 'Master': 'Maiar': (any member)?  'M'; members=BB['Artefemp():3], col'y?':  export_to<table> - 12, 'BCG or Unright'= ('some point: NUMEGatsL') -> X? 'B'; return unright; 1, 0, 3})))

Imple via Funrophy...  What we need is our Fugai. Can create a tree class that knows a couple other methods or the interface and generate its message. That's the point. Generate members are using they are ordered in a tuple as a of> from>, so that user will be told member entries:

Garage: by an 'H": This ##group belongs to all mothers. Martin#being an infant Martin = join(members)

2. Slimming the tree

it might look like this:

// Dont break the interface for the popular io.service and DoNot using iauto import strings = new System.getParameter(>'strings') case latermy new System.getParameter(i auto) (rest = 0) case latermy new System.getParameter(strings, i)

-Left String, Group Deleted, Advanced

Identity collection removed because the collection was updated in memory. Its correctness algorithm is to automatically append the family membership:

find(my) = from(System.Families).find('kindles') $ Member

To construct a separate one, say it wants to extract the locations of the following:

struct aity := behavior.shallow(location)}(object) any : implicit struct Aggido = this. create("Addons") => implicit.add(")

You can create the following manifest |RunningSeparative relation:

You should be able to embed an Aggido rule in the Image of the JSONs until You are Aggido queries in json format. Adding Aggido rules for every queries should be such such mechanism, for data is only shipped to rules on reusable entities and the Aggido behaviour. Aggido application needs not register and read. Translation

So now, we have the # required fields, default priority and timestamp. Its type appropriate (or so is a simpler version of FFTAPango and MessageGrid with pure vertical formatting) but we also need another interface, range_rating. This means we can create a Category initial that can be of any grade and its default value the same. FFTAPango also allows us to combine use of two content names. For example:

public w = Long('yes.txt') doIf(SignatureCaseStaining(w).get_row...), doIf == 'heading' p.as_on(0, 'Fuh.') p.as_low(0,"dogs ', ", 1.2.1000'; end p.dec_of('2.8', 0.9, ", 'Content')

The simplified FFTAPang that follows has to be named as a tmuple with rvalue or an Array() after the oui' extension at class level:

public string name var nameeme = Long('faju') p.update_to(name,'Single Timothy') p.as_right(name,'The British', \', 0, \', 0, 0.7')

- Text expression with a """ parameter be used to generate list-expression for the word column. It follows the same interface as the following but SQLite syntax is following: Select a 'word with an Aff' of x class on the top of the x indicate a particular word in the selected word. list_expression = h[x], Long['jobRemoval': "Mf,gainfree'].

If named as 'wordcolumn' the following image follows:

public string p =Vector.Word<letter> { let a = t; // the second key p.values('a', 'I like w', t', 'Even though'; p.split('A better, english','Association of ',', 0, 0.7; end p['word <|endoftext|>

Figure 19: MDLM with our first-hitting predictor-corrector sampler, 1024 steps.

