# OpenReview forum: "Masked Diffusion Models are Secretly Time-Agnostic Masked Models and Exploit Inaccurate Categorical Sampling"
_ICLR.cc/2025/Conference — ICLR 2025 Poster_

### Official Review · Reviewer_o25N · 2024-10-31

**Soundness:** 4
**Presentation:** 4
**Contribution:** 3
**Rating:** 6
**Confidence:** 4

**Summary:**

The paper investigates Masked Diffusion Models (MDMs), which have gained popularity for discrete generative tasks, particularly for language modeling, where they are now competitive with auto-regressive models (ARMs). Recent work has simplified MDMs by aligning them with continuous-space diffusion models, improving training and sampling strategies. However, the authors reveal a key insight: MDMs’ training and sampling processes do not fundamentally rely on time variables (a typical diffusion model feature) and are instead equivalent to masked models. This finding is illustrated through their proposed "first-hitting sampler" (FHS), which mirrors MDMs' original sampling process but is up to 20 times faster by avoiding time-intensive categorical sampling.

The paper also challenges the assumption that MDMs outperform ARMs in text generation. It identifies a numerical limitation, even with 32-bit floating-point precision, that leads to inaccurate categorical sampling. This flaw reduces the effective temperature, thereby lowering token diversity in generated text. Consequently, the authors suggest that previous quality evaluations using perplexity metrics alone may not accurately reflect MDMs' performance.

**Strengths:**

1.The paper uncovers MDMs' time-agnostic properties, showing they align closely with masked models, simplifying their conceptual and practical applications.
2.The First-Hitting Sampler (FHS) reduces categorical sampling inefficiencies, achieving a 20x speedup, beneficial for real-time applications.
3.Addressing the 32-bit Gumbel sampling precision issue highlights the authors’ rigor, demonstrating 64-bit sampling better preserves entropy and token diversity.
4.Standardizing precision and sampling provides a fair comparison with ARMs, challenging prior MDM superiority claims in generative tasks.

**Weaknesses:**

1.While the authors advocate for MDMs' applications in order-agnostic settings, their comparative analysis still favors ARMs in text generation tasks, possibly limiting the generalizability of the claims for broader applications.
2.The proposed First-Hitting Sampler and high-order sampling extensions add a layer of complexity that might limit accessibility for practitioners less versed in advanced diffusion techniques
3.The experiments primarily focus on language tasks and lack extensive cross-domain testing (e.g., images, audio), where MDMs might exhibit different performance dynamics, which limits the scope of applicability

**Questions:**

1.Could the authors provide additional insight into how MDMs might perform in non-text generation tasks, specifically in visual or audio data domains?
2.How sensitive is the First-Hitting Sampler to different vocabulary sizes or sequence lengths? Would the efficiency gains remain consistent across a wider range of data?
3.Are there scenarios where 32-bit sampling could be advantageous, considering computational resources, or is 64-bit sampling universally superior for maintaining token diversity?

---

> ### Author Response · Authors · 2024-11-21
>
> Thank you for your positive comments and for considering our paper for acceptance. Below, we provide detailed responses to your concerns.
>
> > The proposed First-Hitting Sampler and high-order sampling extensions add a layer of complexity that might limit accessibility for practitioners less versed in advanced diffusion techniques
>
> We would like to explain that the first-hitting sampler is quite simple. It is like a token-by-token decoding process of masked models: each step, uniformly choose a token among the masked positions, and unmask it by sampling from the class probabilities given by the network at this position. Our sampler is only distinct in that, there is a continuous time variable input to the network in MDMs. We update each token's time by Equation 11, which is also very simple.
>
> As for the high-order extensions, they do look complex. But in practice, they are not very necessary as the improvement compared to the first-order case is not as large as in diffusion models.
>
>
> > Could the authors provide additional insight into how MDMs might perform in non-text generation tasks, specifically in visual or audio data domains?
>
> We believe MDM can outperform ARM in other data domains. As we said in the conclusion, "Despite our negative findings, we acknowledge that our text-based experiments may inherently favor ARMs, as text naturally follows a left-to-right order that ARMs are better suited to model. We believe that MDMs are potentially well-suited for applications where the data’s order-agnostic nature is a key prior.". Actually, the evidence can already be revealed by combining our work and recent works. For example, MAR[1] is a masked model built on continuous tokens. They conduct ablations to demonstrate that masked model>ARM in their setting. As our work prove that MDM=masked model, it can be concluded that MDM>ARM in the image domain.
>
> > How sensitive is the First-Hitting Sampler to different vocabulary sizes or sequence lengths? Would the efficiency gains remain consistent across a wider range of data?
>
> Great question! Due to the theoretical equivalence to MDM's original sampling and the connection to masked models's sampling, the correctness of the FHS is guaranteed and irrelevant to vocabulary sizes or sequence lengths. However, the efficiency gains (measured by inference wall-clock time) can depend on many factors and may not as large as 20x in the paper case.
>
> Denote the sequence length as $L$, vocabulary size as $|V|$, the number of sampling steps (for original MDM sampling) as $N$, the number of function evaluations as $NFE$, and the number of categorical sampling operations as $NCS$. As stated in Section 4.1, compared to the caching strategy, the FHS reduces the inference time by reducing $NCS$.
>
> - For original MDM sampling with caching, $NFE\approx N(1-(1-1/N)^L)$, $NCS=NL|V|$.
> - For FHS, $NCS=L|V|$.
>
> The time cost is $NFE\times t_1+NCS\times t_2$, where
>
> - $t_1$ is the time for one network calling, and is related to model size
> - $t_2$ is the time for categorical sampling averaged on 1 position and 1 class, which involves 2 log operations and is fixed.
>
> For fair comparison, we need to compare under the same $NFE$, so that the generation quality is similar. The inference time ratio will be $(NFE\times t_1+NL|V|t_2):(NFE\times t_1+L|V|t_2)$. Therefore, the FHS will give a larger speed-up ratio when:
>
> - The model size is smaller, so that $t_1$ is smaller, and the categorical sampling is relatively more expensive
> - $NFE$ or $L,|V|$ are larger, so that $NCS$ is larger for original MDM sampling
>
> For example:
> - In the paper case, $|V|=50526,L=1024$. To match the original MDM sampling at $N=10000$ steps, we have $NFE\approx 973$. Besides, the model size is around 600M, and $t_1:L|V|t_2\approx 1:1.56$. In this case, the inference time ratio is about 17, around 20x speed-up. If $N=2048$, the ratio will be around 5.
> - DiffSound[2] has 3 times larger model and a much smaller vocabulary. Specifically, $L=265,|V|=256$, and $t_1:L|V|t_2\approx 14.6:1$. Therefore, the categorical sampling is relatively much cheaper. If we also use fewer steps $N=100$, then $NFE\approx 93$, and the speed-up ratio is only about 1.07.
>
>
> > Are there scenarios where 32-bit sampling could be advantageous, considering computational resources, or is 64-bit sampling universally superior for maintaining token diversity?
>
> In fact, with our first-hitting sampler, we don't need 64-bit sampling. Our first-hitting sampler provides a token-by-token decoding process for MDMs. We find that, token-by-token decoding processes (like the decoding process of ARMs) do not suffer from notable numerical issues under 32-bit. We provide explanations and illustrations in Appendix I.2.2 in our revised paper, and welcome reading.
>
> [1] Autoregressive Image Generation without Vector Quantization
>
> [2] Diffsound: Discrete Diffusion Model for Text-to-sound Generation
>
> Thank you again for your consideration, and we are happy to answer further questions.

---

### Official Review · Reviewer_KDgp · 2024-11-03

**Soundness:** 3
**Presentation:** 2
**Contribution:** 3
**Rating:** 8
**Confidence:** 3

**Summary:**

This work provides a deeper understanding of the recently proposed masked diffusion models for discrete generation. It reveals three issues regarding MDM:

1. A key issue is that MDMs's training is free of the time variable and likely learns as time-agnostic masked models. The NELBO objective for training can be reparametrized as time-independent. According to proposition 3.2, the optimal MDM is irrelevant to time. I did not check the seemingly correct proof in detail, but this could be a valuable observation regarding the limitations of MDMs.

2. The sota sampling strategies are time-consuming. Instead, this paper proposes a first-hitting sampler that can achieve better efficiency.

3. There is a numerical fault regarding the 32-bit Gumbel sampling, causing the previous evaluations to be unfair (positively biased) for the MDM. This work offers a fair evaluation and reveals that MDM cannot effectively model discrete generations, such as texts, compared to autoregressive models.

Disclaimer: My review may change if other reviewers identify any problematic issues in the proofs and I have validated them.

**Strengths:**

Novel and recent research problem. MDMs have become popular since 2023, but the theoretical foundations are relatively overlooked. This paper presents key theoretical insights regarding the training and sampling of MDMs.

Technically solid claims and high potential impact. The proofs in the appendix look good to me. The parametrization of training MDMs using NELBO objective to be time-agnostic could facilitate further research on the fundamentals of MDMs.

Some of the claims are validated with experiment results on the main pages, and a re-evaluation of MDMs is presented.

**Weaknesses:**

As there are multiple claims and they seem to be disconnected, I felt it hard to follow sometimes, especially when the topic shifts from training to sampling. As there is much content presented in this paper, I would suggest having a paragraph in the introduction commenting on the organization and flow of this paper.

This paper does not offer detailed experimental validation of the proposed first-hitting sampler compared to existing sampling strategies, like the mentioned works on caching strategies, line 263.

Maybe more experiments can be conducted on the generation quality of different MDM training strategies in Appendix I.2.3, but using the same prompts rather than the current version. Additionally, there seems to be no analysis with a link to I.2.3 anywhere in the main texts or appendix.

I have no idea how to reproduce the results from the paper, and I don't see any supplementary material.

Minor:
1. Figure 1 is a bit ambiguous. You prove MDM = Masked Model, but MDM falls into the Discrete Diffusion region.
2. I was confused when looking at the title. The "Secretly" in this paper title actually marks a negative result on MDM, but for the reader who knows the source from the DPO paper last year [1], it is easy for them to interpret it as a positive result. Additionally, the numerical fault is not included. I cannot come up with any better candidate, to be honest, but I would suggest reconsidering the paper title.

[1] Direct Preference Optimization: Your Language Model is Secretly a Reward Model. NeurIPS 2023.

**Questions:**

1. Can you comment on the limitations of this research and its potential impact on society as well as the research community?

2. Is it possible to mitigate the time-agnostic issue by re-designing or regulating the training objective?

I think this paper offers adequate and solid theoretical insights, and I am inclined to give it a rating of 8. However, I am not entirely confident and would appreciate it if you could address the weaknesses I have mentioned. Thank you.

---

> ### Author Response · Authors · 2024-11-21
>
> Thank you for your positive comments and for considering our paper for acceptance. Below, we provide detailed responses to your concerns.
>
> > As there are multiple claims and they seem to be disconnected, I felt it hard to follow sometimes, especially when the topic shifts from training to sampling. As there is much content presented in this paper, I would suggest having a paragraph in the introduction commenting on the organization and flow of this paper.
>
> We fully understand your concern. Our initial motivation is to study the nature of MDMs in essence, as we feel that MDMs are at the intersection of diffusion models/masked models and lack enough understanding. On a high level, our contribution is to **unify two types of generative models for discrete data: MDMs (discrete diffusion) and masked models**. **If two paradigms of probabilistic generative models are equivalent in both training and inference, then they are the same model.** That is the reason why we demonstrate from these two aspects. We think we have tried to organize and highlight our study in the introduction, and we would like to give more explanations here.
>
> What we highlight in our introduction is (1) unify MDM and masked model, in training and inference (2) uncover that previous evaluation (based on Gen PPL metric) is not comprehensive, and is hacked by lower temperature caused by hidden numerical issues. We believe they all serve towards the goal of an ultimate and thorough understanding of MDMs. **Then we present 3 paragraphs, which start with "for training" "for sampling" "for evaluation", respectively. They correspond to the 3 sections in the main text.** We are unsure whether adding an extra paragraph saying "section xx talks about xx..." is possible, due to the 10-page limitation.
>
> We would appreciate it if you have suggestions for better organization.
>
> > This paper does not offer detailed experimental validation of the proposed first-hitting sampler compared to existing sampling strategies, like the mentioned works on caching strategies, line 263.
>
> We think we have experimented with our sampler, in comparison with other sampling strategies in Section 6, Figure 9.
>
> > Maybe more experiments can be conducted on the generation quality of different MDM training strategies in Appendix I.2.3, but using the same prompts rather than the current version. Additionally, there seems to be no analysis with a link to I.2.3 anywhere in the main texts or appendix.
>
> Thank you for your suggestion! We did not comprehensively test the generation quality of different MDM training strategies, but in rough testing they also perform similarly in sampling. We have added a reference to Appendix I.2.3 in the revised paper.
>
> > I have no idea how to reproduce the results from the paper, and I don't see any supplementary material.
>
> We think our contributions are mostly theoretical, and the main experimental advances are (1) the first-hitting sampler and (2) the numerical issue. We want to explain that they require minimal engineering to reproduce.
>
> For the first-hitting sampler, consider the first-order case in Algorithm 1. Actually, it is very simple. It is like a token-by-token decoding process of masked models: each step, uniformly choose a token among the masked positions, and unmask it by sampling from the class probabilities given by the network at this position. Our sampler is only distinct in that, it handles a continuous time variable which is input to the network in MDMs. Each step's corresponding time is updated by Equation 11, which is also very simple.
>
> For the numerical issue, Figure 8 has given illustrations. We just need to add `dtype=torch.float64` in `torch.rand(...)` function in the categorical sampling part in open-sourced diffusion language model codebases like SEDD and MDLM.
>
>
> > Figure 1 is a bit ambiguous. You prove MDM = Masked Model, but MDM falls into the Discrete Diffusion region.
>
> What we want to express is that, the three-circle part is what the mainstream, or previous works think: MDM is the absorbing and best-performing case in discrete diffusion models. The lower text (MDM = Masked Model) is how our work provides something new: we break previous cognition and unify the two paradigms.

---

> > ### Author Response · Authors · 2024-11-21
> >
> > > I was confused when looking at the title. The "Secretly" in this paper title actually marks a negative result on MDM, but for the reader who knows the source from the DPO paper last year [1], it is easy for them to interpret it as a positive result. Additionally, the numerical fault is not included. I cannot come up with any better candidate, to be honest, but I would suggest reconsidering the paper title.
> >
> > Thank you for your consideration. We agree that a proper title is difficult to craft for this paper, and we also put effort into it. We want to clarify that we have included the numerical fault in the title by "exploit inaccurate categorical sampling". We think "exploit" can already express some negative meanings. "exploit", on the negative side, means "to take unfair or unethical advantage of someone or something for personal benefit": the numerical inaccuracy secretly benefits the incomplete Gen PPL metric, and make people misjudge the advantage of MDMs on text.
> >
> > > Can you comment on the limitations of this research and its potential impact on society as well as the research community?
> >
> > The limitation is that, our investigations are mostly negative. We did not bring improvements to MDMs, such as network architecture modifications, instead negated MDMs on text. We think our research tells people that "MDMs can be abandoned, as it is equivalent to the masked models while being more complex in formulations and slower in sampling. We can directly use masked models in the future." As MDMs are the best-performing discrete diffusion models, our research may further question the potential of the broader discrete diffusion family.
> >
> > > Is it possible to mitigate the time-agnostic issue by re-designing or regulating the training objective?
> >
> > Actually, we don't think "time-agnostic" is an issue. As we said in the last response, we believe it is not promising to consider "how to rescue MDMs". Instead, masked models (like BERT, MaskGIT) are equivalent and simpler, so why not directly use masked models? Recent works like MAR[1] have proven that masked models can outperform ARMs in the image domain.
> >
> > [1] Autoregressive Image Generation without Vector Quantization
> >
> >
> > Thank you again for your consideration, and for giving a positive rating even if not entirely sure. We hope that our response can resolve your concerns, and we are happy to answer further questions.

---

> ### Comment · Reviewer_KDgp · 2024-11-28
> **Thanks for your clarifications**
>
> I appreciate the clarifications from the authors.
>
> > Multiple seemingly disconnected claims.
>
> **I think reviewer nW6z also has this confusion**. I understand that the primary goal of this manuscript is to unify MDMs and masked models while presenting a critical challenge to MDM research. While multiple claims is not a fatal issue for the paper—given its multiple contributions—it is an observation I had while reading. Sharing this feedback is intended to help improve the paper’s readability and, ultimately, its impact. This research will be more beneficial if MDM researchers can easily, fully engage with the paper and clearly grasp its main points.
>
> I am not requesting a detailed breakdown such as "section xx talks about xx," as I believe this is equivalent to the current writing and does not add much value. What I think would be more useful is a focus on the logical connections between the arguments. For instance, why introduce a new sampler for MDMs if the main argument is to abandon them?
>
> A clearer and more logically structured expression, in my opinion, would look something like this:
> "We argue that the community should reconsider investing efforts in MDMs due to the following reasons:
> (1) The training objective of MDMs is essentially the same as that of masked models, as we prove in this paper.
> (2) The sampling process of MDMs is computationally expensive and inefficient.
> (3) The previously reported positive results on MDMs stem from numerical issues during evaluation, rather than genuine advantages."
>
> This kind of framing ties the arguments together cohesively and emphasizes the reasoning behind the paper's contributions.
>
> > Experimental validation of the proposed first-hitting sampler
>
> Thanks for the clarifications. **Could you please further clarify why the generative perplexity (fig 9(a)) differs a lot between the MDMs and AR? if they are theoretically equivalent, shouldn't their performance be similar?**
>
> > Generation quality
>
> Appendix I.2.3 looks good to me now. Thanks for your further efforts.
>
> > Reproducibility
>
> Thank you for providing additional details about reproducibility during the rebuttal. I acknowledge that the primary contribution of this work is theoretical, and the implementation for this manuscript is relatively straightforward. However, I still encourage releasing any source code used to produce the paper results in the future to ensure transparency and facilitate the integrity of the research.
>
> **Minors (which doesn't affect my evaluation)**
> > Figure 1
>
> The key is that the figure shows what "previous works think". Please include the explanations in the paper because there are other interpretations (for example, the one I had in my initial review).
>
> > Title
>
> Thank you for your detailed explanations. I understand that summarizing all the contributions of this paper in a concise title is not a trivial task. The current title is reasonable, although it feels slightly counterintuitive to me. That said, I am unable to think of a better alternative at the moment.
>
> > Limitations
>
> Thanks for elaborating on the limitations. They make sense to me, and please include it in the camera-ready (it doesn't count towards the 10-page limit).
>
> > Improve MDMs.
>
> **I think reviewer Z3xj also has this comment**. I referred to both the response to me and reviewer Z3xj. I appreciate your efforts for trying to improve MDMs but failed and I acknowledge the existing contributions in the paper.
>
> In general, showing that discrete diffusion models and masked models are the same, which has a high potential impact, is both novel and technical. The rebuttal addressed most of my confusion. Therefore, I would lean towards an acceptance of this manuscript, which is a good paper in my opinion. My concern is that **I did not check all the proofs in detail (and it seems none of the other reviewers explicitly mentioned they have verified all proofs)**, therefore I would increase my rating to 8 but keep a confidence of 3 **if the authors could address my final confusion in "Experimental validation of the proposed first-hitting sampler"**. Still, I would suggest the authors to reconsider about the writing.
>
> Disclaimer: My review may change if other reviewers identify any problematic issues in the proofs and I have validated them.

---

> > ### Author Response · Authors · 2024-11-29
> > **Thank you**
> >
> > Thank you for your highly detailed feedback! We highly appreciate your effort in reviewing our work, which will greatly help us improve.
> >
> > > For instance, why introduce a new sampler for MDMs if the main argument is to abandon them? A clearer and more logically structured expression, in my opinion, would look something like this.
> >
> > We believe introducing a new sampler is the preliminary for simplifying MDMs to masked models. In people's understanding, MDMs are more theoretically grounded as it is a well-defined probabilistic model. FHS serves as a bridge, equivalent to the principled sampling of MDMs, and establishes connections to the token-by-token decoding of masked models. Besides, FHS is more general, as it can be applied to the MDM models with time-dependent networks. People may still train and use time-dependent MDMs, following the discrete diffusion works (such as SEDD, the ICML2024 best paper).
> >
> > We highly appreciate your suggestions for reorganizing the introduction! Your example clearly shows more motivations and highlights the connections. We will use your writing with some modifications to replace our paragraph. We may change the second point to "The sampling process of MDMs, being computationally expensive and inefficient, has a theoretically equivalent
> > while more efficient procedure that resembles masked models".
> >
> > > Could you please further clarify why the generative perplexity (fig 9(a)) differs a lot between the MDMs and AR? if they are theoretically equivalent, shouldn't their performance be similar?
> >
> > While MDMs and ARs all possess a token-by-token decoding process, they are theoretically different in the order. ARs follow a fixed left-to-right order, while MDMs (or masked models) randomly choose a position to unmask, so it is in random order. This is like the difference between GPT and BERT. In both training and sampling, ARs are playing "next-token-prediction", while MDMs are playing "filling in the blank". Therefore, the modeling space of MDMs is significantly larger than ARs, making MDMs harder to learn well. In text generation, where a left-to-right order naturally exists, ARs fit this prior well. As verified by GPT and BERT, ARs are more suitable for "text generation", while MDMs are more suitable for "language understanding".
> >
> > > Reproducibility. Include explicit figure explanations/limitations
> >
> > We'll release our code upon acceptance. We will add "previous works think" in the figure and include the limitations in the camera-ready version.
> >
> > Thank you again for your great engagement in the discussion! We are happy to answer further questions.

---

> ### Comment · Reviewer_KDgp · 2024-11-29
> **Thanks for your further clarifications**
>
> I appreciate further clarification from the authors regarding why the performance gap happens between two models that have proven to be equivalent. From my perspective, this is a good paper in general. The rebuttal has effectively addressed my initial concerns, and as a result, I am inclined to raise my rating from 6 to 8.
>
> I sincerely thank the authors for their dedicated efforts in both the research and the detailed rebuttal. Please incorporate the details discussed during the rebuttal phase into the camera-ready version for improved clarity and completeness.

---

> > ### Author Response · Authors · 2024-11-29
> > **Thank you**
> >
> > We are glad that our responses addressed your concerns! Thank you for recognizing our efforts and providing constructive suggestions, which we will incorporate into the final revision.

---

### Official Review · Reviewer_Z3xj · 2024-11-03

**Soundness:** 4
**Presentation:** 4
**Contribution:** 3
**Rating:** 8
**Confidence:** 3

**Summary:**

In this paper, the authors revealed theoretical essence of MDMs, including: 1) MDMs, in both training and sampling, are essentially time-agnostic masked models. 2) MDMs could be significantly lagging behind ARMs in generative perplexity.

**Strengths:**

1.	The paper provides a comprehensive theoretical analysis of Masked Diffusion Models (MDMs), revealing that both their training and sampling processes are effectively time-agnostic and equivalent to masked models. The theory is novel.
2.	The authors introduce the First-Hitting Sampler, a novel sampling method that is theoretically equivalent to the original MDM sampling process but significantly more efficient, enhancing MDM's computational efficiency.
3.	The structure of the article is well organized, with detailed proofs and a thorough analysis of the core ideas.

**Weaknesses:**

1.	The paper shows that MDMs do not outperform ARMs in text generation. It would be beneficial to propose improvements for MDMs.
2.	The experiments are only conducted on text generation; more discrete data generation should be considered. Image generation could also be extended to discrete diffusion models and other discrete data like music generation.

**Questions:**

1.	Have you considered modifications to the model architecture that might help close the performance gap with ARMS?
2.	Will MDM outperform ARMs on other discrete data like music generation?

---

> ### Author Response · Authors · 2024-11-21
>
> Thank you for your positive comments and for considering our paper for acceptance. Below, we provide detailed responses to your concerns.
>
> > The paper shows that MDMs do not outperform ARMs in text generation. It would be beneficial to propose improvements for MDMs. Have you considered modifications to the model architecture that might help close the performance gap with ARMS?
>
> Thank you for your suggestion! We attempted to improve the training through some other techniques (flow matching/variance reduction/self-conditioning, appendix I.1.2), despite they failed. As for the model architecture, we did not modify it because (1) the current one is a quite modern one (DiT+rotaty positional embedding), which is suitable for scaling up the network size. We think in the current era, making local architecture modifications is not as important, and people are more likely to stack simple and popular blocks that have proven scalable. (2) Architecture modification may not close the performance gap with ARMs, as ARMs can adopt the same architecture if there is a better one. We believe it is more important to distinguish which "generative paradigm" is more suitable for certain domain.
>
> > The experiments are only conducted on text generation; more discrete data generation should be considered. Will MDM outperform ARMs on other discrete data like music generation?
>
> Thank you for your suggestion! We only consider text generation as it is the main application of MDMs people are concerned about, the only setting in works like SEDD (ICML2024 best paper) and MDLM (NeurIPS2024). We believe our findings can be naturally extended to other data domains.
>
> We also believe MDM can outperform ARM in other data domains. As we said in the conclusion, "Despite our negative findings, we acknowledge that our text-based experiments may inherently favor ARMs, as text naturally follows a left-to-right order that ARMs are better suited to model. We believe that MDMs are potentially well-suited for applications where the data’s order-agnostic nature is a key prior.". Actually, the evidence can already be revealed by combining our work and recent works. For example, MAR[1] is a masked model built on continuous tokens. They conduct ablations to demonstrate that masked model>ARM in their setting. As our work proves that MDM=masked model, it can be concluded that MDM>ARM in the image domain. Nevertheless, we think a better way is to just abandon MDM, and instead use the equivalent while simpler masked models.
>
> [1] Autoregressive Image Generation without Vector Quantization
>
> Thank you again for your consideration, and we are happy to answer further questions.

---

> > ### Comment · Reviewer_Z3xj · 2024-11-27
> > **Thanks for the response**
> >
> > I prefer to keep my rating.

---

> > > ### Author Response · Authors · 2024-11-27
> > > **Thank you**
> > >
> > > Thank you for taking the time to review our response and for your thoughtful evaluation of our work. We appreciate your support and are open to additional questions or suggestions

---

### Official Review · Reviewer_nW6z · 2024-11-04

**Soundness:** 3
**Presentation:** 2
**Contribution:** 3
**Rating:** 6
**Confidence:** 2

**Summary:**

This paper has two primary objectives. First, it draws a connection between masked diffusion models and time-agnostic; second, the paper examines various strategies for diffusion model training and examines implications of their choices (in particular, caching, samping techniques, and choice of floating point precision).

**Strengths:**

There are aspects of the paper I perceive to be strengths.

For example, empirical evaluations seem by and large well-designed.

The writing is clear; points are argued; there seem to be extensive details in the Appendices.

**Weaknesses:**

There are aspects of the paper I perceive to be weaknesses, or at least invitations for further discussion.

Parts of the text read more like a text book - useful, but I am left wanting to see the implications drawn out. For example, some claims could be elaborated upon, and might be hard for readers to get. For example, the mixture of experts claims on p. 4 has significance not clearly outlined (why does the observation matter, other than as an observation?).

The paper is also not particularly "tight" in the sense of capturing one primary contribution. It is an investigation into a range of phenomena associated with the training of masked diffusion models. The investigation is also very different in its theoretical exploration and its empirical exploration (with the two not really depending on each other much, at least in the float32 discussion). This might make the paper's contribution a bit difficult for readers to find or grasp.

Because at least a part of the paper's contributions are about numerical issues, some experiments about how the resulting issues affect performance in programs with other backends could be instructive for the reader (e.g., JAX, MLX). Some of the numerical precision analysis is of course theoretical but the interaction of the multiple layers of approximation and discretization could play out very differently in different settings.

Overall, the work appears solid, but is seems to be weakened by what could seen as lack of connection between different parts of the paper.

**Questions:**

What do you think is the reason that "we find the truncation under 32-bit precision not influential in the token-by-token decoding process of ARMs."?

If float32 truncation effectively reduces the temperature of the Gumbel, would comparisons between appropriately temperature set ARM and MDM models be fair?

**Details Of Ethics Concerns:**

No ethical concerns.

---

> ### Author Response · Authors · 2024-11-21
>
> Thank you for your constructive suggestions. Below, we provide detailed responses to your concerns.
>
> > For example, some claims could be elaborated upon, and might be hard for readers to get. For example, the mixture of experts claims on p. 4 has significance not clearly outlined (why does the observation matter, other than as an observation?).
>
>
> Thank you for your suggestion. Through the mixture of expert analyses, what we want to express is that the "time-dependent" network implicitly gives a "time-independent" network via aggregation, while *the aggregation is concentrated near the masked ratio*. This gives some intuition why the continuous time is not necessary, as *the masked ratio is discrete, and can be directly obtained from the sequence*. In our training experiments in Appendix I.1, when we alter the network to condition on the masked ratio instead of the continuous time, the training result is similar. We have revised our paper to add a reference.
>
>
> > The paper is also not particularly "tight" in the sense of capturing one primary contribution. It is an investigation into a range of phenomena associated with the training of masked diffusion models. The investigation is also very different in its theoretical exploration and its empirical exploration (with the two not really depending on each other much, at least in the float32 discussion). This might make the paper's contribution a bit difficult for readers to find or grasp.
>
> We fully understand your concern. Our initial motivation is to study the nature of MDMs in essence, as we feel that MDMs are at the intersection of diffusion models/masked models and lack enough understanding. On a high level, our contribution is to **unify two types of generative models for discrete data: MDMs (discrete diffusion) and masked models**. This is conducted through two aspects: training and inference (sampling). If two probabilistic generative models are equivalent in both training and inference, then they are the same model. Therefore, **we respectively disagree that "It is an investigation into a range of phenomena associated with the training of masked diffusion models", as the sampler and the numerical issues are all associated with inference instead of training.** For training, we prove an equivalence of continuous and discrete training objectives. For sampling, our sampler recovers the token-by-token process of masked models, except that we additionally handle the time variable for time-dependent MDMs. In both aspects, we show that there are many complications and misunderstandings of MDMs. We believe the conclusion that MDMs are essentially masked models (like BERT, with some technical differences) can facilitate the researchers, as masked models are simpler in training and faster in sampling, an easier model to use.
>
>
> As for the numerical issue, it is revealed during our experiments. We empirically find that our theoretically equivalent sampler cannot match the original sampling under 32-bit (as the token-by-token process does not suffer much from numerical issues). The original sampling tends to produce lower generative perplexity (which is used in previous works, such as SEDD, the ICML2024 best paper, to advertise the advantage of MDMs). **It is a quite hard and valuable finding that, the generative perplexity metric is not comprehensive, and the diversity is reduced due to very hidden and tricky numerical issues.** We believe the numerical issue is quite a novel and surprising finding to the researchers and forms an essential part of our unification process.
>
>
> > some experiments about how the resulting issues affect performance in programs with other backends could be instructive for the reader (e.g., JAX, MLX). Some of the numerical precision analysis is of course theoretical but the interaction of the multiple layers of approximation and discretization could play out very differently in different settings.
>
> We are not sure what "multiple layers of approximation and discretization" refers to. We want to clarify that, the numerical issue only concerns a very simple mathematical problem: sampling from a categorical distribution (Equation 10 in our paper). It is irrelevant to the network architecture, model training or precision in other places. The parameters of the categorical distribution (i.e., the class probabilities, which are affected by the network output) are all the same for different sampling procedures, and only the simplest procedure, i.e., how to conduct categorical sampling from these given class probabilities, is different and affected by precision. Therefore, the numerical issue is independent of different backends or pretrained models.

---

> > ### Author Response · Authors · 2024-11-21
> >
> > > What do you think is the reason that "we find the truncation under 32-bit precision not influential in the token-by-token decoding process of ARMs."?
> >
> > Great question, and thanks for carefully reading our paper to notice this remark! Actually, our first-hitting sampler is also a token-by-token decoding process (for MDMs), except that we additionally handle the time variable. We also find it does not suffer from numerical issues under 32-bit. We give some explanations and illustrations in Appendix I.2.2 in our revised paper, and welcome to read them.
> >
> > > If float32 truncation effectively reduces the temperature of the Gumbel, would comparisons between appropriately temperature set ARM and MDM models be fair?
> >
> > Sure! We already do so in Figure 10, by comparing MDMs (original sampling), ARMs (+temperature) and MDMs (first-hitting sampler, +temperature). We believe comparing the "Gen PPL-Entropy" trade-off curve is fair: comparing Gen PPL under the same Entropy or vice versa. According to Figure 10, a fair comparison shows that MDMs are not as good as ARMs.
> >
> > Thank you again for your suggestions to help improve our work. We can understand your concerns, and we hope our responses can help you get the ideas we want to express in this paper, and the connection between different parts (they all serve towards the goal of an ultimate and thorough understanding of MDMs). If you have further questions, we are happy to discuss.

---

> > > ### Comment · Reviewer_nW6z · 2024-11-25
> > > **Thanks to the authors for these clarifications**
> > >
> > > Thanks to the authors for these detailed clarifications. They are very helpful. I have revised my score favorably.
> > >
> > > I see some of the connections drawn better - whatever the authors can do in revisions to explicitly spell out connections between the various insights contained in the paper as contributing towards their holistic conclusion, the better.

---

> > > > ### Author Response · Authors · 2024-11-25
> > > >
> > > > Thanks for your positive feedback! We are glad to hear that our responses are helpful. We agree that explicitly spelling out connections is better, and we will make revisions depending on the 10-page limit.

---

### Meta-Review · Area_Chair_y7To · 2024-12-23

**Metareview:**

The paper provide a theoretical analysis of the recently proposed masked diffusion models (MDMs) for discrete generation, showing that MDMs are essentially time-agnostic masked models. Further, the paper also introduces a novel sampling method that is theoretically equivalent to the original MDM sampling process but significantly more efficient. Finally, the paper also validates their sampling method on text generation. All the reviewers were positive about the contribution, highlighting its clarity, potential impact and timeliness. They did provided a few points for improvement that the reviewers addressed during the rebuttal period. As a consequence, I recommend acceptance.

**Additional Comments On Reviewer Discussion:**

The authors put a significant effort in addressing the reviewers' concerns during the rebuttal period and several reviewers did increase their overall rating persuaded by the authors' arguments.

---

### Decision · Program_Chairs · 2025-01-22

Accept (Poster)